# Interleaving asynchronous and synchronous activity in balanced cortical networks with short term synaptic depression

Jeffrey B. Dunworth[1,2,10], Yunlong Xu[3,4,5,10], Michael Graupner[6], Bard Ermentrout [1,7], Alex D. Reyes[8] & Brent Doiron [1,4,5,7,9] ✉

Cortical populations are in a broadly asynchronous state that is sporadically interrupted by brief epochs of coordinated population activity. Inhibitory stabilized networks reproduce a low activity asynchronous regime but cannot generate population events. In contrast, synaptic depression stabilized excitatory networks create transient surges of activity, yet give inhibition only a perfunctory role. We analyzed spontaneously active in vitro mouse auditory cortex slices that show both regimes, including slow (2–12 Hz) oscillations within some events. We built firing rate and biophysically realistic spiking models in which excitation is balanced by recurrent inhibition, yet all excitatory synapses undergo short term depression. In our model a depression of synaptic excitation onto inhibition neurons initiates events, while depression of excitation onto excitatory neurons shapes rhythmicity of the events, reproducing the full repertoire observed experimentally. Our work unifies balanced and depression stabilized theories and provides a mechanistic framework for nonlinear, population wide correlations in cortex.

Internally generated cortical activity is a reflection of the circuit structure and physiology of the cortical network[1–3]. Circuit models provide an important tool to test and validate whether a specific biological feature of cortex can mechanistically explain recorded population dynamics[4–6]. However, it is often the case that cortical models are built to capture only a subset of cortical dynamics. Indeed, there are distinct recurrent circuit models for asynchronous population dynamics[7,8], rhythmic synchrony[9], long timescale dynamics[10], and population-wide coordinated behavior[11–13]. In many cases, these models assume only the circuit structure and physiology needed to replicate the population activity of interest, while ignoring the biology that is critical in the other models. One clear reason for this multiplicity of cortical models is that real cortical dynamics are quite rich, with

activity that transitions between distinct states throughout a recording, and models tend to focus only upon a single dynamical state.

For the majority of the time, cortex is in an asynchronous (or weakly correlated) state with temporally irregular spiking dynamics[8,14,15] and population responses fluctuate with linear dynamics about an operating point[2,5,16]. Models with strong recurrent excitation and inhibition readily capture these dynamics[6,17–21], and they produce fluctuations in population activity that are Gaussian distributed[8]. However, the asynchronous dynamics in biological cortical networks are sporadically interrupted by brief epochs of population-wide coordinated activity, as reported in spontaneously active in vitro slice recordings[22–25], spontaneously active cortex in anesthetized animals[26–28], and even in the cortices of awake behaving rodents[29] and

[1]Department of Mathematics, University of Pittsburgh, Pittsburgh, PA, USA. [2]Department of Mathematics, University of Michigan, Ann Arbor, MI, USA. [3]Graduate Program in Computational Neuroscience, University of Chicago, Chicago, IL, USA. [4]Grossman Center for Quantitative Biology and Human Behavior, University of Chicago, Chicago, IL, USA. [5]Department of Neurobiology, University of Chicago, Chicago, IL, USA. [6]Université Paris Cité, CNRS, Saints-Pères Paris Institute for the Neurosciences, Paris, France. [7]Center for the Neural Basis of Cognition, Pittsburgh, PA, USA. [8]Center for Neural Science, New York University, New York, NY, USA. [9]Department of Statistics, University of Chicago, Chicago, IL, USA. [10]These authors contributed equally: Jeffrey B. Dunworth, Yunlong Xu. ✉e-mail: bdoiron@uchicago.edu

primates[30]. These epochs of large population correlation are clear nonlinear phenomena and cannot be captured by classical network models where inhibition and excitation are balanced[8,18]. Circuit models that do capture these population events often require a brief respite from short-term synaptic depression of recurrent excitation that drives population activity[12,31–33]. This mechanism ignores the wealth of data suggesting that inhibition confers strong stability to population activity[25,28,34–37]. The apparent disconnect between models where population stability is due to strong recurrent inhibition and those where nonlinear runaway population activity emerges through a relief from synaptic depression represents a clear barrier towards a more complete mechanistic theory of cortical dynamics.

In this study, we focus on the internally generated dynamics of spontaneously active in vitro slices of rodent auditory cortex[25]. Whole cell patch recordings show that the cortical network is largely in a state where recurrent inhibition tracks and balances excitation, except for rare and short periods when the slice undergoes a population-wide surge in activity. A subset of the slice preparations show a population burst of activity with a previously unreported low frequency (2–12 Hz) within-burst rhythm, while the remaining preparations show population events that lack any rhythm. We present novel analysis and modeling of the complex and varied nature of the population events. To capture this dataset, we follow past work[38] and extend balanced network models to include synaptic depression from excitatory neurons to both excitatory and inhibitory neurons. This gives a balanced asynchronous solution where fluctuations in activity may cause a depression-induced weakening of inhibitory recruitment, prompting the excitatory population to explode into a population event. Further, the rhythmic character of population events results from an interplay between recurrent excitation and the depression of excitation. This feature requires that excitatory synapses onto inhibitory neurons exhibit more depression at relatively lower firing rates as compared to depression onto excitatory cells. Fortunately, this requirement has strong support from past electrophysiological recordings in primary auditory cortex[39–41]. We show that this distinction between synaptic plasticity can be captured by a significant synaptic facilitation for excitation onto excitatory neurons, giving a biological basis for our key requirement.

It is clear that the cortex exhibits diverse dynamics poised between asynchrony and population-wide correlations. Our framework gives a clear blueprint for how to combine and extend past models so as to account for previously unexplained cortical dynamics.

## Results

### Two kinds of network stabilization

We first revisit previously published recordings from layer IV neurons in an in vitro thalamocortical slice preparation of mouse primary auditory cortex (see ref. 25 for a complete description). An extracellular solution containing high potassium, low magnesium, and low calcium concentrations was used to recruit internally generated neuronal activity within the slice[42]. Whole-cell voltage clamp techniques isolated the excitatory currents into a single neuron by holding its membrane potential at approximately the reversal potential of inhibition (−80 mV). The spontaneous network activity typically provided a net excitation which fluctuated about a low state, suggestive of asynchrony between the pre-synaptic excitatory neurons (Fig. 1a). However, this state was sporadically interrupted by large excursions in excitation lasting hundreds of milliseconds, presumably due to coordinated recruitment of excitatory neurons throughout the network (Fig. 1a, b). Similar population dynamics have been reported in the spontaneous states of cultured[31,43], in vitro[24,44], and in vivo[29,45] preparations. We label these rare epochs population events, similar in nature to the "population spikes" and "population bursts" described in past studies[12,32]. In particular, population event dynamics are reported in auditory cortex during

in vivo spontaneous conditions, determined either directly from population recordings[27] or indirectly from whole cell recordings that show large membrane potential excursions suggestive of widespread synchrony[26].

Population event dynamics are captured by networks of spiking neuron models having recurrent excitatory connections with short-term synaptic depression[12,31–33,46]. The activity of these models reduces to a simple dynamical system with only excitatory activity and synaptic depression as variables[32,33,47,48], making them amenable to phase plane analysis (Fig. 1c). The positive feedback associated with recurrent excitation could in principle destabilize network activity; however, strong synaptic depression weakens recurrent coupling so that moderate network activity is stable (Fig. 1c, intersection of orange and red curves). Consequently, we will throughout refer to these networks as depression stabilized. While the asynchronous low activity state is stable, a transient relief from depression promotes a rapid runaway explosion in activity due to strong positive feedback in the excitatory network (Fig. 1c, black curve), modeling the population event. This massive excitatory drive recruits synaptic depression that weakens recurrent feedback, ultimately bringing activity back to the low state and terminating the population event.

Dynamical systems with a single stable state that exhibit large-scale transient excursions in activity as a response to input perturbations above a certain size are often labeled excitable systems[49]. Population event activity in depression stabilized networks is an example of excitable dynamics, and firing rate model reductions of them require two main assumptions. First, the asynchronous spiking dynamics during the periods between population bursts necessitate either sources of external noise that are independent across neurons[31,46,48] or broad neuronal heterogeneities within the population[12,32]. This ensures that the weak recurrent coupling in the depressed state does not synchronize network activity. Second, synaptic depression must operate on a timescale that is much slower than recurrent excitation[47,48], otherwise runaway population activity will be prematurely quenched. While the second assumption is well-founded in synaptic physiology[41,50], the first ignores common input projections that would otherwise synchronize the network[2].

In depression stabilized networks, inhibitory connectivity plays only a perfunctory role; in some models, it is included[12,31,33], while in other models it is completely ignored[32,46–48]. A popular alternative cortical model is one in which strong recurrent inhibition actively counteracts recurrent excitation, effectively stabilizing network activity at a moderate level. Such networks are labeled inhibition stabilized[34,51,52] and include networks with large excitation that is balanced by recurrent inhibition[8,17,18]. In vivo recordings in both visual[21,34,37] and auditory cortices[35,53] have provided strong evidence for an inhibition-stabilized cortex.

Depression and inhibition stabilized networks differ in several key aspects. In inhibition-stabilized networks, the timescale of inhibition must be sufficiently fast to prevent runaway activity. Because of this, inhibition stabilized networks cannot produce a stereotypical population event, and rather produce only stunted transient responses to a slight reduction in inhibition (Fig. 1d). Nevertheless, networks with strong recurrent inhibition require neither sources of independent noise external to the network nor broad cellular heterogeneities to produce asynchronous dynamics; rather, they produce it naturally through inhibitory cancellation of correlations due to common excitation[8,19].

One advantage of spontaneously active in vitro slice preparations is that joint excitatory-inhibitory dynamics can be directly measured[25]. Simultaneous whole-cell recordings from neuron pairs were obtained, where one neuron's membrane potential was held at the inhibitory reversal potential (−80 mV; Fig. 1e, red curve) and the other at the

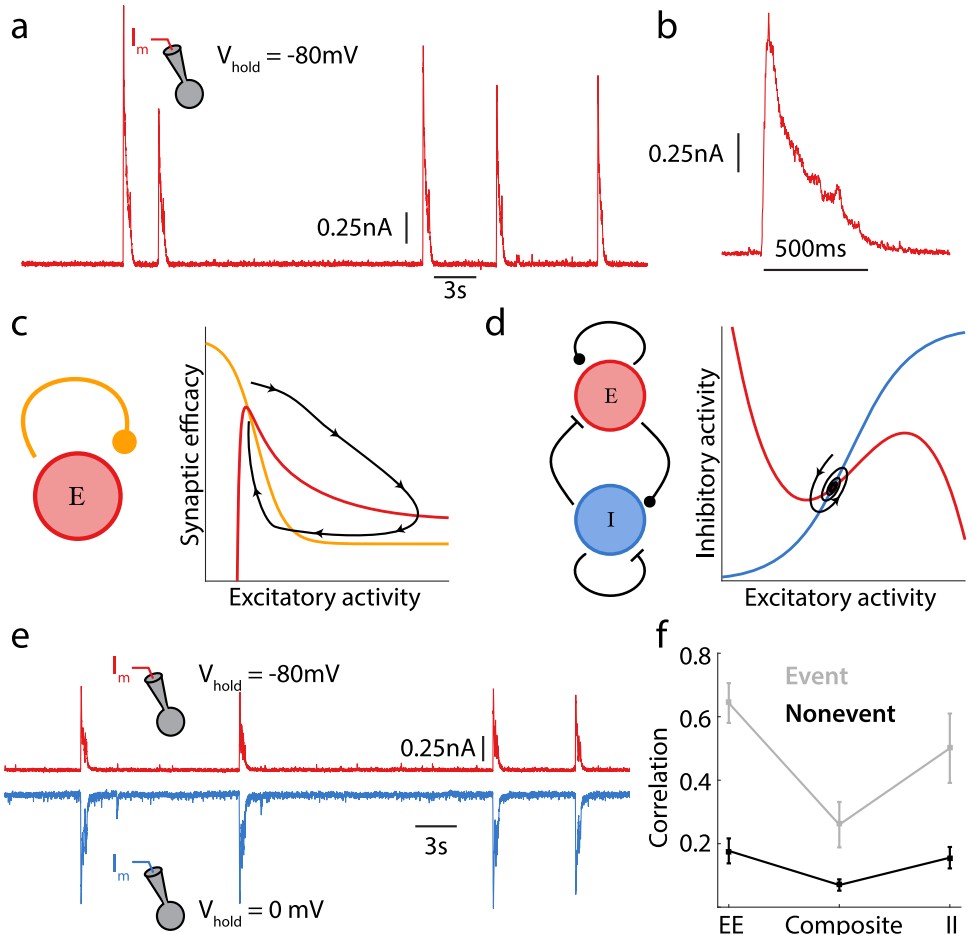

**Fig. 1 | Two mechanisms for stable network activity. a** Example whole cell voltage clamp recording from mouse auditory cortex, in vitro; the neuron is held at the reversal potential for inhibition. **b** Magnification of population event from recording shown in (**a**). **c** Phase portrait for a depression stabilized network; the red and orange curves denote nullclines for excitation and synaptic efficacy (depression), respectively. A momentary release of depression permits large runaway excitatory activity until more depression is recruited, decreasing excitatory activity (black curve). **d** Phase portrait for an inhibitory stabilized network; the red and blue curves denote nullclines for excitation and inhibition, respectively. A perturbation to network activity causes a brief and small amplitude relaxation back to equilibrium (black curve). **e** Synaptic input traces from dual whole cell patch clamp

recordings from mouse auditory cortex. Two neurons are simultaneously clamped, one held at the reversal potential of excitation, the other at the reversal potential of inhibition. **f** The correlation coefficient of synaptic inputs from simultaneously recorded pyramidal ($n_{EE}$ = 17, $n_{Composite}$ = 21, $n_{II}$ = 16; error bars are Standard Error of the Mean). The neuron pair is held at the inhibitory/inhibitory reversal to measure the correlation of excitatory currents (EE), or held at the excitatory/excitatory reversal to measure the correlation of inhibitory currents (II), or held at a membrane potential where both excitatory and inhibitory currents are active to measure the correlation of summed excitation and inhibition (composite). Figure 1 (**a**, **b**, **e**, and **f**) are adapted from Graupner & Reyes[25].

excitatory reversal potential (0 mV; Fig. 1e, blue curve), isolating excitatory and inhibitory population activity, respectively. Restricting analysis of paired neuron activity to the asynchronous period (i.e., excluding population events) shows that while neuron pairs receive both correlated excitation (Fig. 1f, $E-E$) and inhibition (Fig. 1f, $I-I$), the composite input to the pair measured at resting potential is only weakly correlated (Fig. 1f, Composite). This suggests that shared inhibition actively tracks and cancels shared excitation, producing an asynchronous network state as has been previously theorized[8,19], and is a hallmark of inhibitory stabilization.

Taken together, these observations present a problem when considering the spontaneously active in vitro auditory cortex slice data (Fig. 1 and ref. 25). Specifically, the excitatory-inhibitory dynamics recorded between population events support inhibitory stabilized dynamics. Such a cortical model disallows the existence of population event dynamics, and yet population events are clearly present in the data and inhibition is strongly recruited during the excursion (Fig. 1e). The central focus of our study is to provide a circuit model of cortex

that captures the full range of asynchronous and population event dynamics in auditory cortex.

## Oscillatory and non-oscillatory population events

In order to better motivate our model, we first analyze the structure of population activity within events recorded from our spontaneously active cortical slice[25]. We note that any arbitrary pair of recorded cells likely do not share a direct connection, though all patched cells register simultaneous population events throughout the recording. We take this as an indication that the population event is slice-wide, and not likely a recording site artifact.

Broadly speaking, there are two classes of population events: ones where the synaptic excursion is a simple rise and then decay (Fig. 2a, lavender curve) and ones with a low frequency rhythmic dynamic throughout the population event (Fig. 2a, green curve). Even so, population events display a wide range of heterogeneity across all slices (Fig. 2a, black curves). The population events recorded from a given neuron do not switch between these classes: if a neuron showed

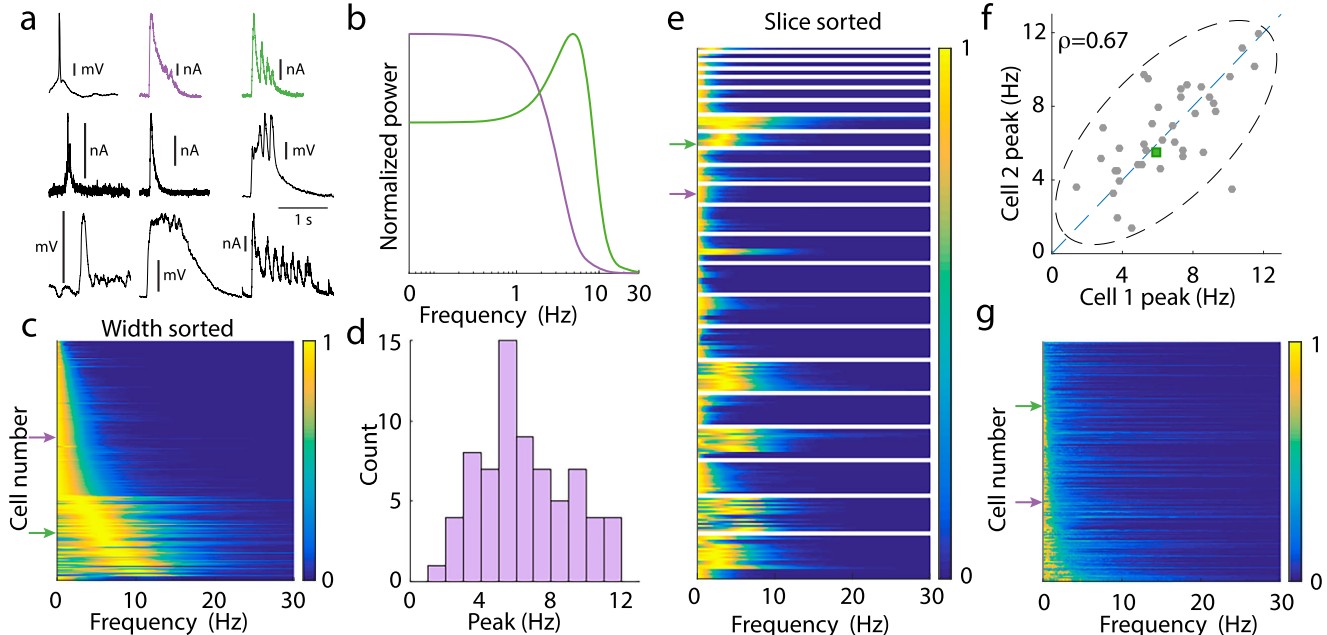

**Fig. 2 | Heterogeneity of population event dynamics. a** Sample population events from different slices. For events recorded in voltage clamp, vertical scale bar denotes 0.2 nA. For events recorded in current clamp, vertical scale bar denotes 10 mV. Sample events marked in lavender (aperiodic) and green (periodic), are exemplar events marked throughout figure. **b** Power spectra for marked cells from the dataset to the left, averaged over all of the events seen. There is a clear peak in the power spectrum, here at ~ 6 Hz. Power spectra are normalized to have maximum value 1. **c** The power spectra for the event periods of the entire dataset. Power spectra are normalized to have maximum value 1. Data sorted by full width at half max (for those without peaks), and then by peak location (for those with peaks). Arrows denote marked data from (**a**). **d** Histogram of the peak location for peaked power spectra. **e** Same data as (**c**), sorted and grouped by slice, rather than by width. **f** Scatterplot of the peak location for pairs of cells. Since some of the collected data measures excitatory currents, some measures inhibitory currents, and some measures mixtures, there is no natural categorization for choosing which cell is "Cell 1" and which is "Cell 2". These labels are chosen at random for this figure. The reported correlation coefficient is the median of the distribution of correlation coefficients computed for different instantiations of the randomization. Dashed ellipse denotes the 95% confidence interval. Periodic data from (**a**) marked as a green square. **g** Same as (**c**) for the nonevent portions of the data.

a rhythmic event during a recording, all events recorded from that neuron would be rhythmic. This is verified by spectral analysis restricted to the population event epochs from these two example neurons, showing that the respective arrhythmic or rhythmic character was recording site specific (Fig. 2b; see "Methods"). Spectral analysis of the population event epochs across the entire dataset ($n = 210$ from 23 distinct slices) shows that approximately one third of recorded neurons exhibited rhythmic events (Fig. 2c). In the oscillatory cases the frequency is relatively slow, and somewhat heterogeneous across the slices, ranging from ~ 2 Hz to 12 Hz (Fig. 2d). A natural question to ask is whether population event dynamics are due to a synaptic and cellular property of the recorded neuron, or a network-wide feature.

Reordering the dataset according to slice membership rather than spectral width shows a slice dependent clustering of oscillatory event dynamics (Fig. 2e). This is consistent with an interpretation of a population event as a network feature, since if one neuron experiences an oscillatory event, then all other neurons in the same slice do as well. Further, the frequency of the oscillation is also highly correlated within a slice; namely, the peaks in the power spectrum for simultaneously recorded neurons lie roughly on the identity line (Fig. 2f). Taken together, these results suggest that the oscillatory character of population events is due to slice heterogeneity, rather than a cellular process that is private to a neuron. This fact will simplify the assumptions of our cortical model.

In contrast to the population event dynamics, the synaptic fluctuations that occurred between population events were dominated by low frequency power, and showed no discernible peak suggestive of rhythmic dynamics (Fig. 2g). A lack of pronounced population-wide oscillations is characteristic of the asynchronous state associated with

the low activity regime[8,19,20]. The fact that about one-third of slices which show oscillatory dynamics during the event do not show it for the low activity state suggests a strong nonlinearity in network dynamics. Overall, these data provide an important constraint—our circuit model must be able to produce both arrhythmic and rhythmic population events, as well as show a clear asynchronous low activity state devoid of any rhythm between population events.

## Balanced cortical circuit model of population events

We first begin with modeling the low activity asynchronous state in which the cortical population resides for most of the time. Motivated by the fact that recurrent inhibition is critical for asynchrony in the auditory cortical slice recordings (Fig. 1f), we consider recurrently coupled populations of excitatory ($E$) and inhibitory ($I$) neurons (Fig. 3a). The excitatory and inhibitory populations have $N_E$ and $N_I$ neurons, respectively. For simplicity, we take $N_E = N_I \equiv N$, but this assumption can be easily relaxed. Since we will be studying network-based mechanisms, we opt for a phenomenological network model that only considers neurons as being active or inactive (we later consider networks of biologically motivated spiking neuron models). We focus on the proportion of active neurons in population $\alpha$, $r_\alpha(t) = n_\alpha(t)/N$, where $n_\alpha(t)$ is the number of active neurons at time $t$ (here $\alpha \in \{E, I\}$).

Following classical work in the theory of balanced excitation and inhibition[8,18], we scale the baseline synaptic strength from a neuron in population $\beta$ to a neuron in population $\alpha$ as $J_{\alpha\beta} = j_{\alpha\beta}/\sqrt{K}$, where $j_{\alpha\beta} \sim \mathcal{O}(1)$ is the unscaled connection strength, and $K$ is the average number of connections onto a neuron. Here we will take $K = cN$, with $c \in (0, 1]$ being a fixed and positive number. This scaling for $J_{\alpha\beta}$ implies very strong connections between neurons, especially when compared to

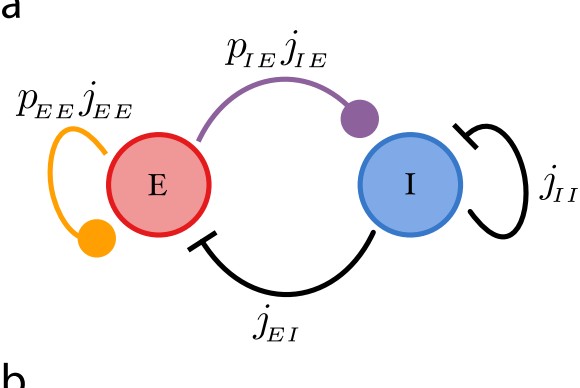

a

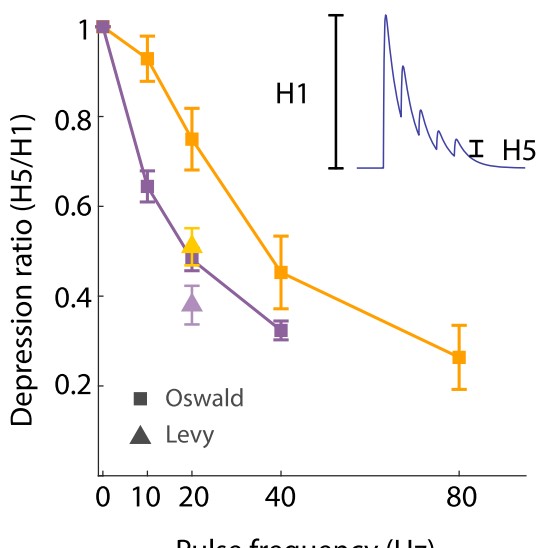

b

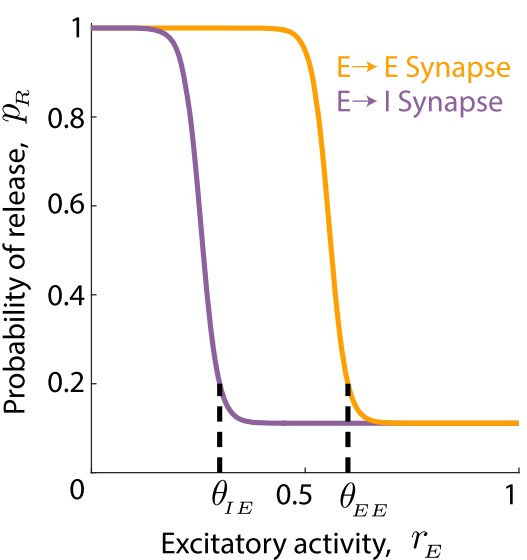

c

**Fig. 3 | Balanced network model with synaptic depression of excitatory connections. a** Schematic of the full rate model. Excitatory (red) and inhibitory (blue) populations are modeled as Markov processes, while the dynamical synapse variables ($p_{EE}$, $p_{IE}$) are modeled as continuous, slowly varying variables. Connection strengths ($j_{\alpha\beta}$) are marked. Dynamical variables are all scaled to live in [0, 1]. **b** Data from Oswald & Reyes[39] and Levy & Reyes[57] showing amount of depression as a function of pulse frequency. Depression is measured as the ratio of the response heights from the fifth spike to the first spike (see *Inset*). Synapses between pyramidal cells are shown in orange: *n* = 34 (10 Hz), 25 (20 Hz), 23 (40 Hz), 12 (80 Hz). Synapses from pyramidal to inhibitory shown in purple: *n* = 78 (20 Hz). Symbols give mean values and error bars are ± SEM across recorded pairs. **c** Release probability ($p_R$) as a function of excitatory firing rate ($r_E$) used in the model. Threshold parameters $\theta_{\alpha\beta}$ are marked. The only difference between the curves is the threshold for the onset of depression. To agree with the data in (**b**), the threshold for the EE synapse is chosen as larger than the threshold for the IE synapse ($\theta_{EE} > \theta_{IE}$).

We take a sigmoidal activation function $f_\alpha$ for population $\alpha$, and for large $K$ the population activity then obeys the following dynamics (see "Methods"):

$$\dot{r}_E = -r_E + f_E\left(\sqrt{K}\underbrace{(j_{EE}p_{EE}r_E + j_{EI}r_I + I_E)}_{\text{Balance condition}:\mathcal{O}(1/\sqrt{K})}\right),$$

$$\tau_I\dot{r}_I = -r_I + f_I\left(\sqrt{K}\underbrace{(j_{IE}p_{IE}r_E + j_{II}r_I + I_I)}_{\text{Balance condition}:\mathcal{O}(1/\sqrt{K})}\right). \qquad (1)$$

Here $\tau_I$ is the time constant of the inhibitory population relative to the excitatory, and $I_\alpha$ is a source of external drive to population $\alpha$. Finally, $p_{\alpha E}$ quantifies the degree of short-term synaptic depression from the $E$ population onto population $\alpha$. To gain intuition for the subtleties of this scaling, for the moment we ignore depression and set $p_{\alpha E} = 1$. The $1/\sqrt{K}$ synaptic scaling introduces a complication, namely that the argument to $f_\alpha$ in principle grows with $\sqrt{K}$. This could produce saturated activity for large $K$, disallowing any reasonable cortical dynamics. However, the network robustly corrects for this through a *balance condition* whereby the combined external and recurrent excitation is roughly canceled by recurrent inhibition so that the total input remains $\mathcal{O}(1)$ for all network sizes. For robust balance conditions, one requires large external drive, i.e., $I_\alpha \sim \mathcal{O}(1)$. Such a balance condition produces asynchronous network dynamics[8], and the dual patch whole cell experiments in ref. 25 (Fig. 1) were originally performed as a validation of this theory.

While balanced network models have been successful at explaining several aspects of cortical spiking dynamics, for $K \to \infty$ the firing rates are nonetheless determined by the pair of linear equations defined by the balanced conditions. In particular, this means that the firing rates ($r_E$, $r_I$) are linearly related to the external inputs ($I_E$, $I_I$), and the nonlinearities of the transfer functions $f_\alpha$ do not contribute to population activity. Additionally, the timescale of cortical dynamics becomes very fast, so that for moderate $\tau_I$ and large $K$ there cannot be rhythmic dynamics[18]. These limitations are serious obstacles when trying to capture the clearly nonlinear population event behavior (Fig. 1), as well as the slow rhythmic dynamics in a subset of the population events (Fig. 2).

Mongillo et al.[38] have extended the theory of balanced networks to include short-term synaptic plasticity. Following their example, we use previous mean field reductions[50,54] to model population-wide synaptic depression, $p_{\alpha E}$, from the $E$ population onto population $\alpha$:

$$\dot{p}_{\alpha E} = \frac{1 - p_{\alpha E}}{\tau_\alpha^r} - \frac{a_\alpha(r_E)p_{\alpha E}}{\tau_\alpha^d}, \qquad (2)$$

those from the more common $1/K$ scaling. As a result, the network produces sizable internally generated variability, so that single neuron activity is temporally irregular, in line with experimental recordings[14]. While $1/\sqrt{K}$ synaptic scaling was originally a theoretical abstraction, recent work in cultured neuronal preparations gives strong evidence for this scaling[43].

## a

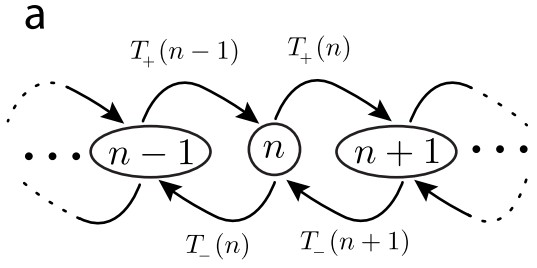

## b

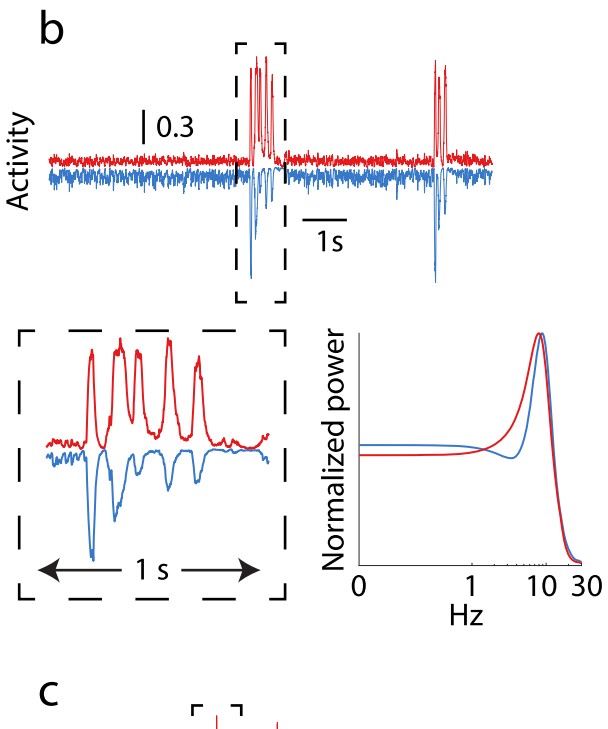

## c

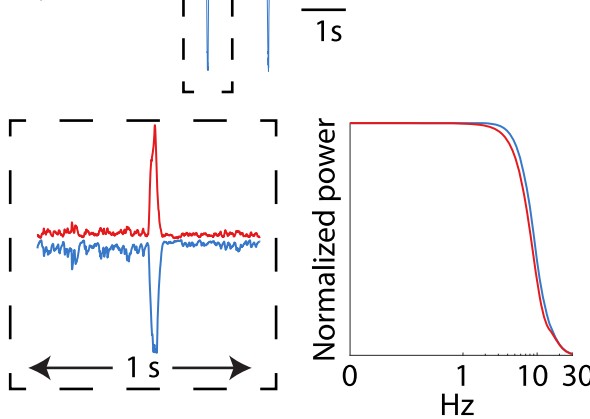

**Fig. 4 | Birth-death Markov model of network activity. a** Each neuron is modeled as a binary neuron that can be in an "active" or "inactive" state, and we consider the number of neurons $n$ in the "active" state. The transition rates between states, $T_+$ and $T_-$, are shown. **b, c** Simulations for two different parameter sets that reproduce the rare, random nature of the events seen in data. In both panels, $K = 400$. **b** These events display an oscillatory nature. A simulation of 10s is shown across the top, while a 1s interval around the first event is shown on the bottom left. Power spectra for the excitatory and inhibitory populations during the event is shown on the bottom right. This simulation shows an oscillation at ~8 Hz. Power spectra are normalized to have maximum value 1. Here $\theta_{EE} = 0.5$, $\theta_{IE} = 0.2$. **c** Same as B, except these events display no oscillatory nature. The power spectrum shows no clear peak, but does have a large degree of low frequency power. Here $\theta_{EE} = 0.5$, $\theta_{IE} = 0.8$.

$$a_\alpha(r_E) = \frac{m_\alpha}{1 + e^{-\beta_\alpha(r_E - \theta_{\alpha E})}}. \qquad (3)$$

The first term on the right-hand side of Eq. (2) models the recovery from synaptic depression, while the second term models its recruitment; these occur on timescales $\tau_\alpha^r$ and $\tau_\alpha^d$, respectively. While

recovery dynamics are simple, recruitment depends nonlinearly on the $E$ population activation through $a_\alpha(r_E)$. This recruitment is sigmoidal in $r_E$ (Eq. (3)), where $\theta_{\alpha E}$ is the threshold level in $r_E$ required for significant depression in the excitatory synapse onto neurons in population $\alpha$. When coupled with the balance conditions in Eq. (1), the short-term depression (STD) dynamics in Eqs. (2) and (3) allow the nonlinearities in $a_\alpha$ to shape the firing rate solutions ($r_E, r_I$). This is because synaptic depression now reduces the baseline connectivity $j_{\alpha E}$ by an $r_E$-dependent solution of Eqs. (2) and (3). Thus, the network firing rates ($r_E, r_I$) must be solved in conjunction with the equilibrium solution of $p_{\alpha E}$, involving strong nonlinearities. For instance, in the original study[38], short-term facilitation of excitatory-excitatory connectivity produced a bistable network with coexisting low and high activity regimes.

STD involves several complicated chemical and biophysical processes[55], all lending to significant heterogeneity of plasticity recruitment and recovery across synapse types[56]. In layer II-III of auditory cortex the STD of both the excitatory to inhibitory and excitatory to excitatory synapses has been measured through simultaneous in vitro patch clamp recording[39,40] (the results are summarized in Fig. 3b). Here, it is clear that $\theta_{IE} < \theta_{EE}$, meaning that the recruitment of depression of the $E \rightarrow I$ synaptic pathway occurs at lower $r_E$ than depression of the $E \rightarrow E$ pathway (compare purple vs. orange curves in Fig. 3b). Similar recordings from layer IV of auditory cortex[57] are consistent with the overall trend we observe (Fig. 3b, triangles). We adopt this depression heterogeneity in our model (Fig. 3c); this will be shown this to be a critical requirement to capture the full range of the population behavior recorded in the auditory cortical slice. We remark that for simplicity our model ignores any short-term plasticity (STP) of the inhibitory connectivity, despite these connections being shown to depress in auditory cortex[40,58].

In the large network size limit ($K \rightarrow \infty$), the dynamics of Eqs. (1)–(3) lack any stochastic aspect to their population activity. This means that solutions cannot capture the random activation time of population bursts, that is a clear feature of the in vitro data. The collective behavior of finitely many neurons often introduces new stochastic effects at the population level that are absent in theories that take $K \rightarrow \infty$[20,59–63]. To explore our model with finite $K$ we study the associated birth-death Markov process whose large $K$ limit is the model in Eq. (1)[61]. Briefly, the number of active neurons $n_\alpha$ undergoes stochastic transitions to either $n_\alpha + 1$ (birth) or $n_\alpha - 1$ (death) (Fig. 4a; see "Methods"). The stochastic birth and death of population activity imparts a random character to $r_\alpha(t)$ that is shaped by the recurrent excitatory/inhibitory circuit dynamics along with the slower synaptic depression. Simulations of the finite size cortical network show fluctuating population activity about a low state, punctuated by sporadic population events (Fig. 4b, c, top). Depending on the depression thresholds $\theta_{EE}$ and $\theta_{IE}$ the population events show either rhythmic (Fig. 4b, bottom) or arrhythmic (Fig. 4c, bottom) dynamics. Our analysis reveals that as the number of neurons in the network decreases, stochastic fluctuations increase, leading to more frequent population events. This behavior

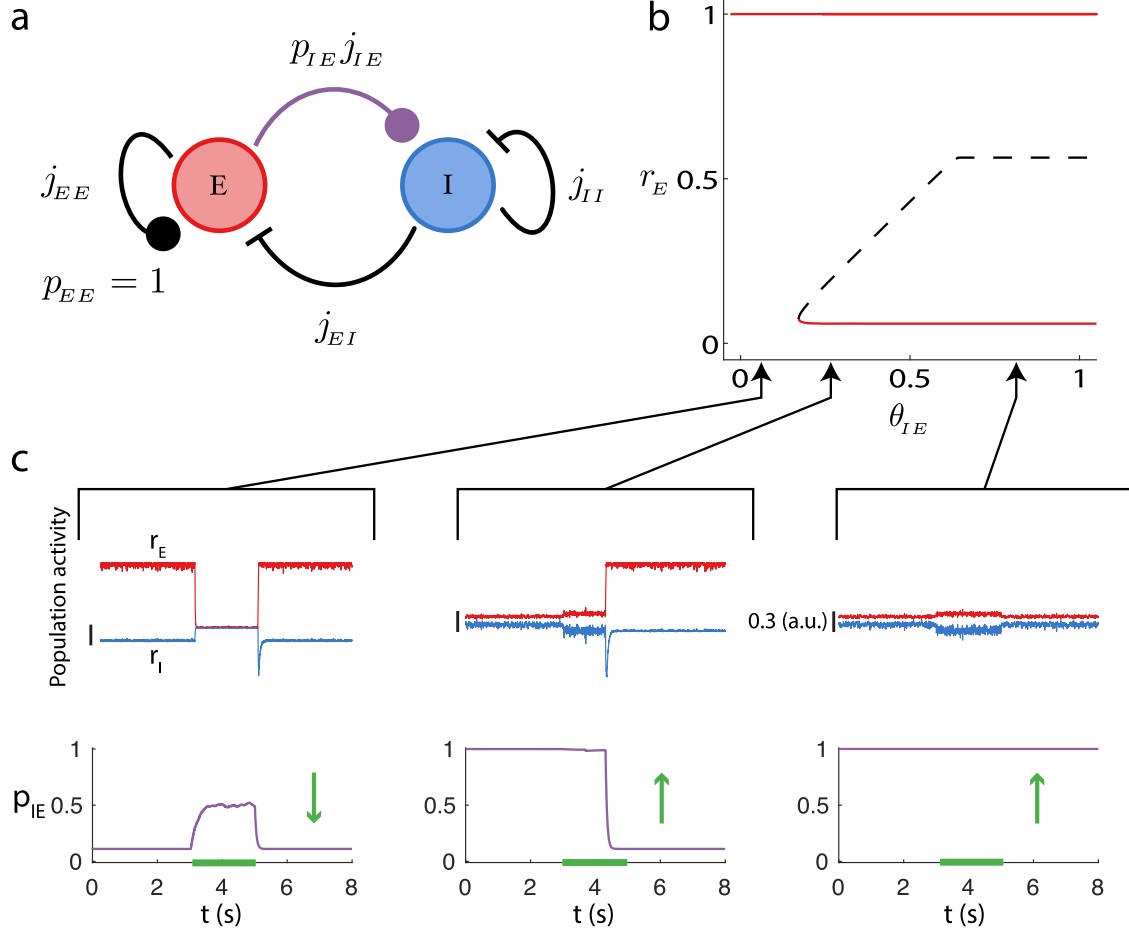

**Fig. 5 | Reduced network model with fixed $E \to E$ synaptic strength in the undepressed state ($p_{EE} = 1$). a** Schematic of the reduced system. **b** Bifurcation diagram of the excitatory rate ($r_E$) as the depression threshold $\theta_{IE}$ is varied. Red lines denote the stable fixed points and the dashed black line denotes an unstable (saddle) fixed point. The lower stable state and saddle point coalesce in a saddle-node bifurcation. For the computation of the bifurcation diagram we took $K = 400$.

**c** Example simulations for three values of $\theta_{IE}$, with $K = 400$. Top row: traces of the activity, $r_E$ in red, $r_I$ in blue. Vertical scale bars denote 0.3. Bottom row: plasticity variable. We applied an external stimulation to the excitatory population from $t = 3$s to $t = 5$s (green bar marks external stimulation). The depression threshold $\theta_{IE}$ is indicated by the black arrows linking to (**b**).

aligns with the theoretical expectations for rare processes and is detailed through the examination of exponentially distributed inter-event intervals that agree with rare event theory[59–63] (see Supplementary section "Finite Size Scaling of Rare Population Events" and Fig. S1).

The simplicity of the model makes it amenable to analysis that reveals how the recurrent excitation and inhibition combine with the two synaptic depression processes to produce rich population event dynamics. In the following sections, we will use the fact that the $E \to I$ synapses depress at lower pre-synaptic rates than $E \to E$ synapses ($\theta_{IE} < \theta_{EE}$; Fig. 3b, c) to probe the separate mechanisms underlying population *event initiation* and *event rhythmicity*.

**Population event initiation**

Population event initiation occurs at $r_E$ values that are far below $\theta_{EE}$, and thus a reasonable approximation for the low activity regime is to take the $E \to E$ synapses as undepressed ($p_{EE} = 1$) (Fig. 5a). This simplification reduces the network dynamics in Eqs. (1)–(3) to three dynamical variables: $r_E(t)$, $r_I(t)$, and $p_{IE}(t)$. While this reduced model cannot replicate the population event dynamics of the full model, its simplicity can give insight on how events are initiated. We perform bifurcation analysis on the reduced system ($r_E$, $r_I$, $p_{IE}$ and use $r_E$ as the indicator of network state (Fig. 5b). In this reduced system, we consider how population activity depends on the threshold for $E \to I$ synaptic

depression, $\theta_{IE}$. For sufficiently low $\theta_{IE}$, so that the $E \to I$ synapse is depressed even at low $r_E$ values, the excitatory population cannot recruit sufficient recurrent inhibition to stabilize a low activity state. Rather, only a stable high activity state exists (Fig. 5b, solid line at $r_E \approx 1$). In this regime, an external (negative) input can transiently lower $r_E(t)$, allowing $p_{IE}(t)$ to recover and $r_I(t)$ to increase despite the reduction in $r_E$ (Fig. 5c, left, 3s < t < 5s). Nevertheless, after stimulus removal, the network quickly returns to the high activity state with $p_{IE}(t)$ returning to very low values (Fig. 5c, left, t > 5s). Thus, cortical models with small $\theta_{IE}$ that do not allow a stable low activity state cannot hope to capture the in vitro population activity (Fig. 1a).

As $\theta_{IE}$ increases a stable low activity state is created (Fig. 5b, solid line at $0 < r_E < < 1$) along with an unstable threshold state (a saddle point) that separates the new low activity regime from the high activity regime (Fig. 5b, dashed line). Here, the dynamics are bistable; when $r_E(t)$ is in the low activity state, a sufficiently strong input can transition the network into the high active state (Fig. 5c, middle, 3s < t < 5s). The transition recruits sufficient $p_{IE}(t)$ depression so that the network remains in the high state after the stimulus is removed (Fig. 5c, middle, t > 5s). This transition from the low to high activity state of $r_E$ is the population event initiation in the full rate model. Finally, for very large $\theta_{IE}$ the network remains bistable, but the saddle point is distant from the low activity state (Fig. 5b; $\theta_{IE} > 0.6$). In effect, raising $\theta_{IE}$ makes the

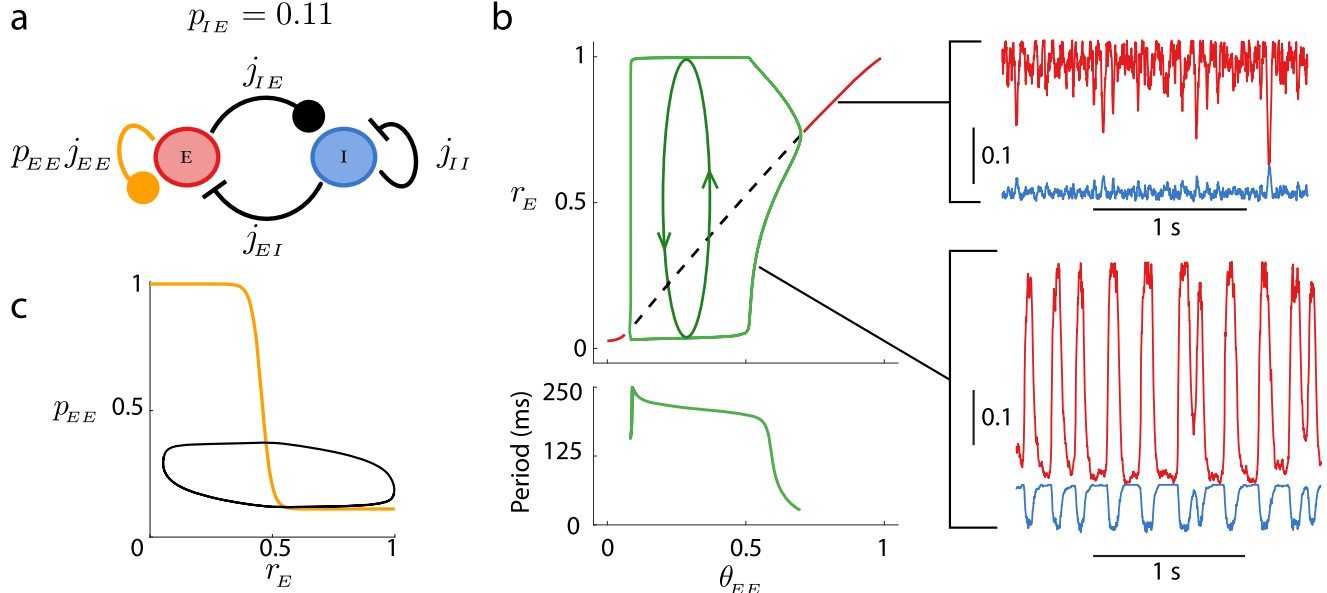

**Fig. 6 | Reduced network model with fixed $E \to I$ synaptic strength in the depressed state ($p_{IE} = 0.11$). a** Schematic of the reduced system. **b** *Top:* Bifurcation diagram of the excitatory rate ($r_E$) as the depression threshold $\theta_{EE}$ is varied. Red lines denote stable fixed points, the dashed black line denotes an unstable fixed point, and the green lines show the peak and trough of a network oscillation (limit cycle). For the computation of the bifurcation diagram, we took $K = 400$. *Bottom:* The period of the oscillation as a function of $\theta_{EE}$. *Right:* Sample simulations for two different values of $\theta_{EE}$ (top: $\theta_{EE} = 0.9$, bottom: $\theta_{EE} = 0.5$), with $K = 400$. **c** Limit cycle of the underlying dynamical system overlaid on the nullcline for $p_{EE}$.

$E \to I$ synapse less depressed, effectively strengthening excitation onto inhibitory neurons. this shifts the inhibitory nullcline so that the saddle drifts farther from the low rate attractor, increasing the fluctuation required to reach it. Therefore, moderate perturbations no longer promote transitions from low to high population activity (Fig. 5c, right). For finite size networks internally generated fluctuations can initiate a population event via a stochastic transition across the stable manifold of the saddle point; these transitions are far more likely for moderate $\theta_{IE}$ where the stable low state and saddle points are near one another.

In sum, the reduced model shows how an activity-dependent weakening of inhibitory recruitment allows a complex dynamic, wherein a low activity inhibitory stabilized state can have a threshold beyond which inhibition can no longer prevent runaway activity from positive feedback. However, our reduced model (with $p_{EE} = 1$) lacks the requisite mechanisms to produce excursions of stereotyped width that are clear in the population events of the full rate model (Fig. 4b, c).

### Population event rhythmicity

To investigate the mechanisms underlying within population event dynamics, we study a reduced model with the $E \to I$ synapses assumed to be depressed ($p_{IE} \approx 0.11$, Fig. 6a), as is the case after event initiation (Fig. 5). This reduction restricts the network dynamics in Eqs. (1)–(3) to three dynamical variables: $r_E(t)$, $r_I(t)$, and $p_{EE}(t)$. In particular, a low $p_{IE}$ value removes the low activity state, and the reduced model is then only appropriate for modeling network dynamics within a population event epoch.

In this reduced model, a large threshold for synaptic depression in the $E \to E$ synapse, $\theta_{EE}$, produces a stable activity state at high $r_E$ (Fig. 6b, solid red curve and top inset). However, as $\theta_{EE}$ is reduced, the recurrent excitation is depressed and the high state loses stability via a supercritical Andronov–Hopf bifurcation. Consequently, this births a stable limit cycle solution of growing amplitude, creating oscillatory dynamics within the network (Fig. 6b, green curves and bottom inset). The oscillation reflects a competitive interplay between $p_{EE}(t)$ and $r_E(t)$, where the oscillatory timescale is set primarily by the depression

recruitment and recovery time constants. Further, over a wide range of $\theta_{EE}$ the frequency is between 4 and 8 Hz (Fig. 6b, bottom), matching the oscillation frequency in the full rate model (Fig. 4b, bottom) and a subset of the in vitro data (Fig. 2c). Overlaying the limit cycle on the plasticity nullcline in the ($p_{EE}$, $r_E$) plane shows how the oscillation samples the full nonlinearity of $p_{EE}$ recruitment (Fig. 6c, cf. Fig. 1c), highlighting the nontrivial interaction between $p_{EE}$ and $r_E$.

The combination of the two reduced models (Fig. 5 and Fig. 6) suggests that the two depression pathways, $p_{IE}$ and $p_{EE}$, play distinct roles in population event initiation and within event dynamics, respectively. Armed with these insights, we next aim to study how population event dynamics depends upon the pair of depression thresholds ($\theta_{EE}$, $\theta_{IE}$) in the full rate model.

### Suite of population event dynamics

To study the range of possible dynamics in the full rate model, we begin by constructing a two-parameter ($\theta_{EE}$, $\theta_{IE}$) bifurcation diagram of Eqs. (1)–(3) (center panel of Fig. 7). There are two main organizing structures in the diagram. First, a curve of saddle-node bifurcations (purple curve in Fig. 7, cf. Fig. 5b) separates regimes with either monostable or bistable dynamics. Bistable dynamics exist for sufficiently large $\theta_{EE}$ and moderate $\theta_{IE}$ (purple region in Fig. 7), while either a low rate solution (yellow region in Fig. 7) or saturated state (gray region in Fig. 7) are possible outside of this region. Second, curves of Andronov–Hopf bifurcations (green curves in Fig. 7, cf. Fig. 6b) demarcate a region with oscillatory dynamics (green region in Fig. 7). The combination of saddle-node and Andronov–Hopf bifurcations provides key intuition into how the network responds to finite $K$ induced internally generated fluctuations.

For sufficiently large $\theta_{IE}$ (so that the system is above both the saddle-node or Andronov–Hopf bifurcations), there is only a stable low state (yellow region in Fig. 7). For low $\theta_{EE}$ the recurrent excitation is very depressed and internally generated fluctuations simply perturb the activity about the low state (upper left inset in Fig. 7). Moderate $\theta_{EE}$ makes it possible for fluctuations to transiently remove depression, resulting in a large population event that is akin to the excitable

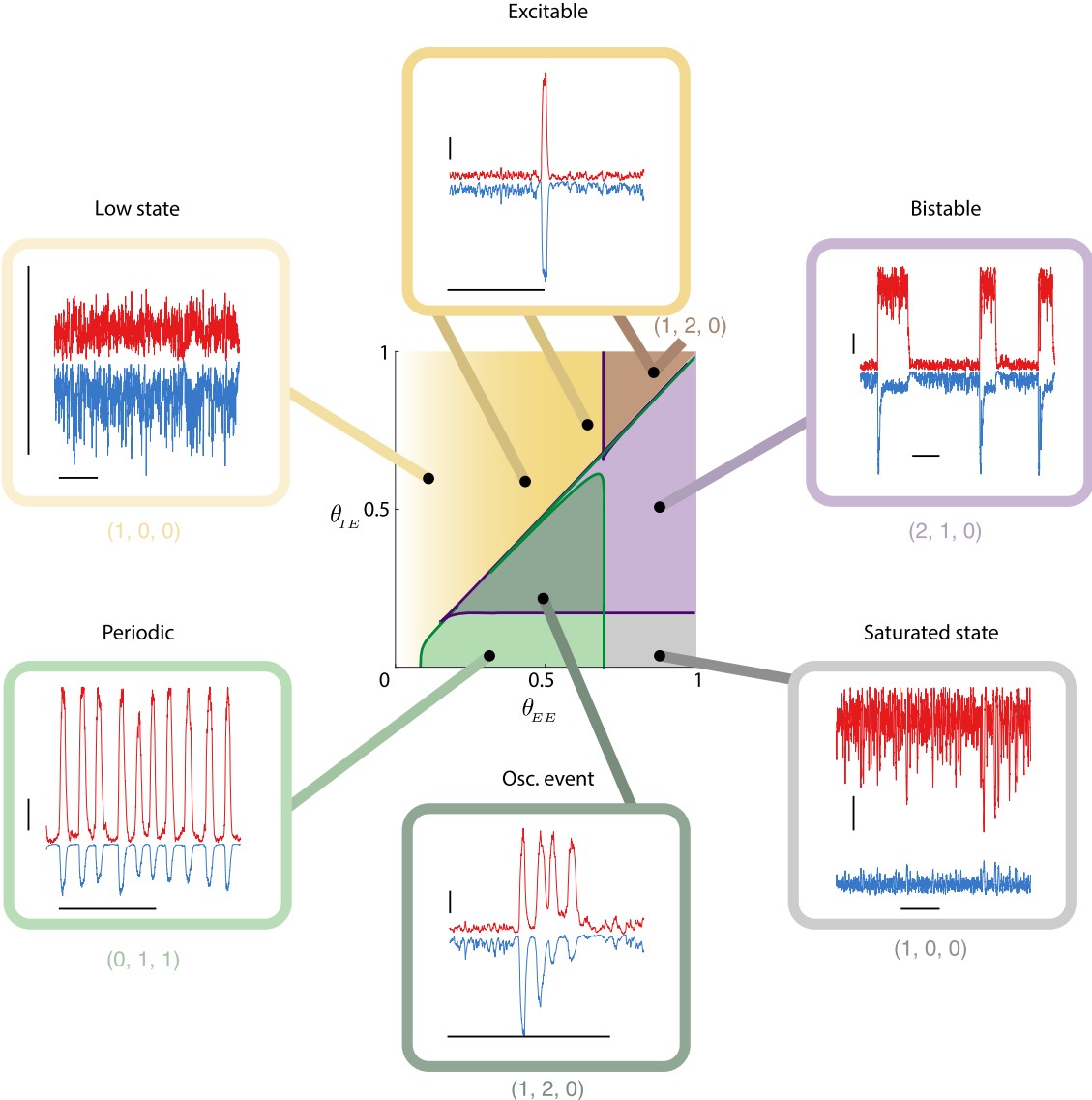

**Fig. 7 | Suite of network dynamics possible in the full rate model.** *Center:* Two parameter bifurcation diagram as the depression thresholds $\theta_{EE}$, $\theta_{IE}$ are varied. The purple lines mark the location of the saddle-node bifurcations (cf. Fig. 5) giving the boundary between monostable and bistable dynamics. To the right of this line (purple shaded regions) the system is bistable. The green line marks the location of a Andronov–Hopf bifurcation (cf. Fig. 6) marking the boundary between non-oscillatory and oscillatory behavior. To the left of the green line but the right of the purple line (green shaded regions), the system can exhibit slow oscillations. Above the unit diagonal, the system is excitable but non-oscillatory (yellow shaded gradient). For low enough values of $\theta_{EE}$ the vector field loses its excitatory nature and

events do not happen. For the computation of the bifurcation diagram, we took $K = 1000$. *Outer ring:* Example simulations showing the suite of possible behaviors. Horizontal bars denote 1s, and vertical bars denote increments of 0.2 in the fraction of active neurons(dimensionless). Simulations are for $K = 400$, and, starting with the bistable region and moving clockwise, the threshold values are $(\theta_{EE}, \theta_{IE}) = \{(0.8, 0.5); (0.8, 0.05); (0.5, 0.2); (0.3, 0.05); (0.1, 0.6); (0.8, 0.9)\}$. The three numbers in tuples below the boxes give the count of (# stable limit points, # unstable limit points, # stable limit cycles) in each region; note that the top-right set corresponds to the top-right dark region.

dynamics explored in past studies[12,31–33,48] (top inset in Fig. 7). However, because the network has not passed through the Andronov–Hopf bifurcation it lacks any oscillatory dynamics and consequently these population events are strictly arrhythmic.

For very low $\theta_{IE}$ and moderate $\theta_{EE}$ the system has passed through the Andronov–Hopf bifurcation and a stable limit cycle solution produces periodic dynamics (bottom left inset in Fig. 7). However, for larger $\theta_{IE}$ the system passes through the saddle-node bifurcation and more subtle population dynamics emerge (dark green region in Fig. 7). In this region, there is no longer a stable limit cycle to organize periodic behavior; rather, there is only a stable low rate state. Nonetheless, finite size fluctuations induce transient population bursts with oscillatory dynamics (bottom inset in Fig. 7). This region of parameter

space contains the ($\theta_{EE}$, $\theta_{IE}$) pairings for the initial simulations of the full rate model (Fig. 4).

Focusing on ($\theta_{EE}$, $\theta_{IE}$) pairings within this region, we find that it is possible to have oscillatory and non-oscillatory population event dynamics. For very low $\theta_{IE}$ (point I in Fig. 8a) there is a globally stable limit cycle solution and an unstable fixed point (phase portrait I in Fig. 8b). Simulations of the full finite size stochastic model show a robust and sustained low frequency oscillation (time series I in Fig. 8b). As $\theta_{IE}$ is increased the system passes through a period doubling bifurcation (red curve in Fig. 8a). This shifts the stable limit cycle to have period two dynamics (II in Fig. 8b); we remark that the limit cycle remains globally stable. As $\theta_{IE}$ is increased further, another period doubling bifurcation (orange curve in Fig. 8a) gives rise to stable

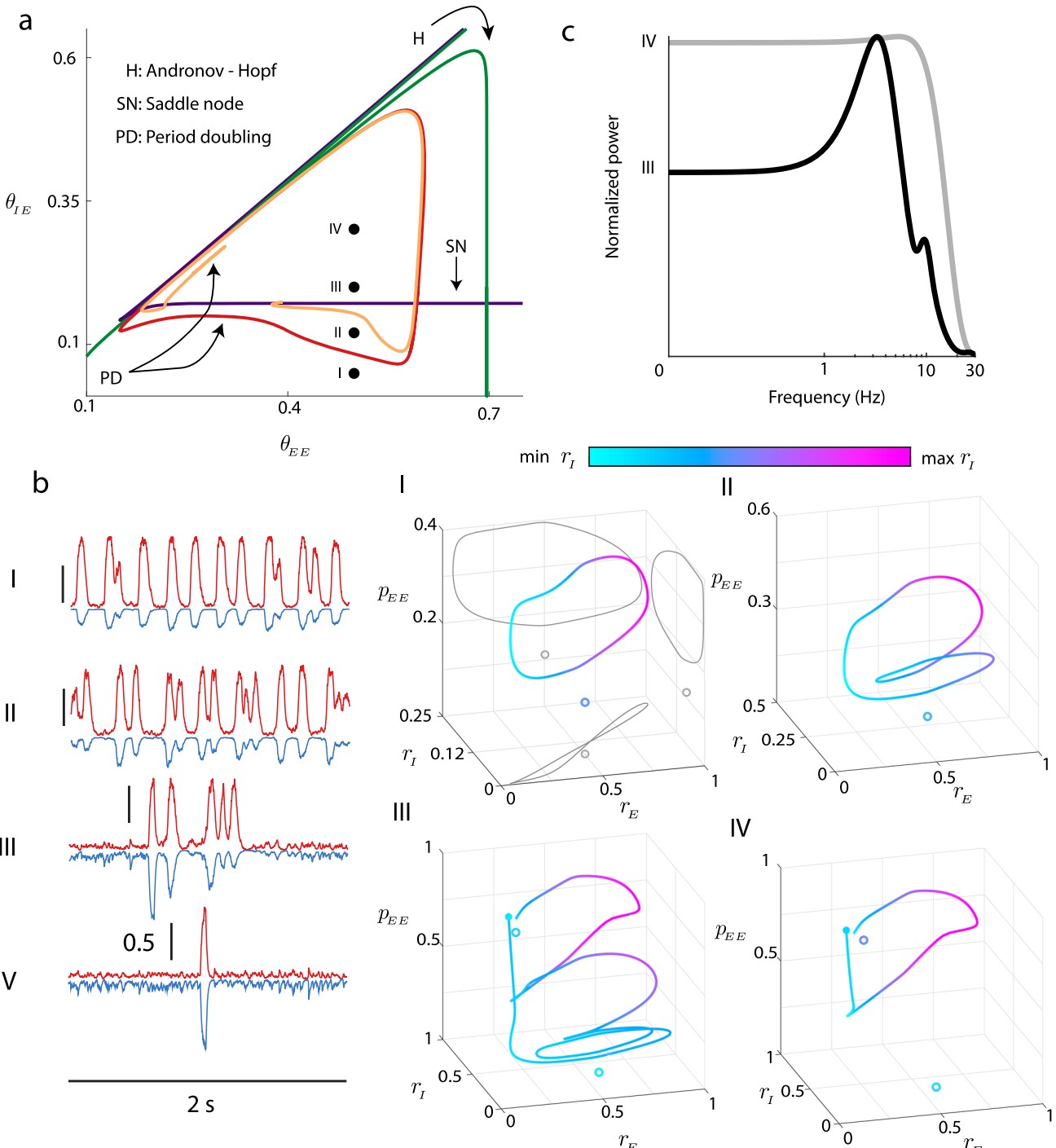

**Fig. 8 | Capturing non-oscillatory and oscillatory population events in the firing rate model. a** Magnification of bifurcation diagram shown in Fig. 7, with period doubling bifurcations added (red/orange curves). Four different parameter choices are marked ($\theta_{EE} = 0.5$, $\theta_{IE} \in \{0.05, 0.12, 0.2, 0.3\}$). **b** *Left:* Sample time courses of noisy system for the four parameter choices. *Right:* Trajectories from deterministic system, displayed in ($r_E$, $r_I$, $p_{EE}$). Color denotes depth (location in $r_I$). Open circles denote unstable fixed points, filled circles denote stable fixed points. For I, we show projections onto each of the coordinate planes (gray curves). For choices I and II, we observe a stable limit cycle. For choices III and IV, after the saddle-node bifurcation, the only attractor is the stable fixed point, but the echo from the limit cycle allows for complex transients in the excitable system. **c** Power spectra for parameter choices III and IV (cf. Fig. 2b).

period four dynamics (not shown). As $\theta_{IE}$ continues to increase, the system undergoes a period doubling cascade, suggesting complicated chaotic transient dynamics. For even larger $\theta_{IE}$ a stable/unstable pair of fixed points are born out of the saddle-node bifurcation (purple curve in Fig. 8a), and the stable limit cycle is lost. Here the full rate model dynamics is excitable, with a slight perturbation from the globally stable fixed point resulting in a large-scale excursion before returning

to the fixed point (phase portrait III in Fig. 8b). However, the high period structure of the stable cycle that existed below the saddle-node bifurcation still shapes the dynamical flow, so that the excursion has a rhythmic character. This is apparent in the fluctuation-induced population event dynamic (time series III in Fig. 8b). Finally, for even larger $\theta_{IE}$ the excitable dynamics persists, but now the excursion is simple and the population event lacks rhythmic dynamics (IV in Fig. 8b). This

is because we are now farther in $(\theta_{EE}, \theta_{IE})$ parameter space from the region where the stable high period limit cycle shaped population dynamics. These last two parameter sets (III and IV in Fig. 8) capture that rhythmic and arrhythmic population event dynamics, respectively; indeed spectral analysis of the time series from the stochastic model (Fig. 8c) reflect that from the in vitro data (Fig. 2b).

### Population events in networks of spiking neuron models with biologically realistic STP

Up to this point, our treatment has focused on modeling synaptic plasticity with the phenomenological model given in Eqs. (2)–(3). This model's simplicity has the distinct advantage of capturing the main distinction between $E \to E$ and $E \to I$ plasticity with parameters $\theta_{IE}$ and $\theta_{EE}$ (Fig. 3). This allowed for the systematic analysis of the onset and oscillatory nature of population events (Figs. 5–8), where we determined that $\theta_{EE} > \theta_{IE}$ is a necessary constraint. However, from a biological perspective, the threshold $\theta$ is not physical or chemical parameter of a synapse, rather it describes the response of a synapse to repeated, periodic inputs. It remains unclear what biological differences between $E \to E$ and $E \to I$ synapses would support $\theta_{EE} > \theta_{IE}$. Further, the rapid onset, rhythmicity and decay of population events are likely influenced by fine timescale spiking dynamics which are absent in firing rate equations of the form Eq. (1). Thus, while our phenomenological model provides tractability, the mechanisms for population events it puts forth may not operate in more biologically realistic frameworks. To address this possibility, we next consider the mechanics of population events in a large-scale network of spiking neuron models where synaptic connections have realistic synaptic STP mechanics (model details are given in the "Methods").

Consider the synapse from pre-synaptic neuron $j$ to postsynaptic neuron $i$, which has synaptic connection strength $J_{ij}(t)$. Following past modeling work[64], we set $J_{ij}(t)$ as the product of a constant scale factor $\bar{J}_{ij}$ and two dynamic factors—a synaptic Depression factor $D(t)$ and a synaptic Facilitation factor $F(t)$:

$$J_{ij}(t) = \bar{J}_{ij}D(t)F(t). \tag{4}$$

Factors $D(t)$ and $F(t)$ are updated after each presynaptic spike and control the dynamical evolution of $J_{ij}(t)$. $D(t)$ is adjusted by multiplying it by a spike-recruited depression update, $d$, modelling the depression (or fatigue) incurred per presynaptic action potential. Similarly, $F(t)$ is incremented by a spike-recruited facilitation update, $f$, reflecting the facilitation per presynaptic spike. Specifically, these rules are given by:

$$D \to D \cdot d, \quad F \to F + f, \quad d \in (0,1], \quad f \in [0, +\infty). \tag{5}$$

Finally, in between pre-synaptic spikes, both $F(t)$ and $D(t)$ recover exponentially back towards a steady value of 1.

For this synaptic model, the parameters $d$ and $f$ define the recruitment and frequency dependence of short-term synaptic plasticity. By setting $d$ and $f$ differently across $E \to I$ and $E \to E$ synapses we can capture an effective synaptic depression of excitation onto $I$ neurons that is recruited at a lower presynaptic rates $r_E$ than for excitation onto $E$ neurons (compare purple and orange curves in Fig. 9a). The distinction is due to the interplay between spike-recruited depression and facilitation being sensitive to the presynaptic firing rate $r_E$.

The $E \to E$ synapse has both synaptic depression ($d = 0.24$) and facilitation ($f = 0.85$). Because of this choice, at low presynaptic $r_E$ any "effective" depression in the $E \to E$ synapses is minimal (Fig. 9b, top plot), due to the factors $D(t)$ and $F(t)$ counteracting one another ($D(t)F(t) \approx 1$; Fig. 9b, bottom plot). However, at higher $r_E$ the $E \to E$ depression factor $D(t)$ dominates over the facilitation factor $F(t)$, and a net effective depression occurs (Fig. 9c). By contrast, the $E \to I$ synapse has only depression ($d = 0.24$ and $f = 0$), so that depression is recruited at low and high presynaptic $r_E$ rates (Fig. 9d, e). Thus, the increased

"threshold" of spike-recruited effective depression in the $E \to E$ synapse compared to the $E \to I$ synapse can be captured by synaptic facilitation in the $E \to E$ synapse.

We incorporate this model of synaptic plasticity within a recurrently coupled network of excitatory ($N_E = 4000$) and inhibitory ($N_I = 1000$) spiking neuron models (see "Methods"). In the baseline (or spontaneous) state the network spiking activity is in low firing rate (mean excitatory rate of 1.13 Hz) and roughly asynchronous regime (Fig. 10a, top), as expected for networks with balanced excitation and inhibition[8]. The spiking network does not endogenously create population events; to do so would require significant fine tuning of network size or connectivity[65,66]. Rather, to investigate the dynamics of population events, we externally stimulated a small fraction of the excitatory neurons (200 neurons with the highest average firing rates) for a duration of 80 ms to initiate population events (Fig. 10a, green boxes). We only consider non-stimulated neurons in our analysis of population dynamics.

In our model, $E \to E$ synapses have both facilitation and depression, making the threshold for "effective" depression high. To explore the dynamical structure of population events we vary the plasticity mechanics of $E \to I$ synapses (mirroring the analysis presented in Fig. 8). If the facilitation update $f_{IE}$ is low (Fig. 10b, purple circle), implying that $E \to I$ synapses have a low threshold to effective depression, then when an event is initiated the population response is oscillatory in both excitatory and inhibitory neurons, with rhythmic dynamics extending well past the stimulation period (Fig. 10a, bottom). By contrast, if $f_{IE}$ is higher (Fig. 10b, green cross) then the event is non-oscillatory and a brief rise in both excitatory and inhibitory activity occurs (Fig. 10c). Finally, if $f_{IE}$ is high and matches $f_{EE}$ (Fig. 10b, gold star) then external stimulation does not result in an event in the excitatory neuron population (Fig. 10d). Spectral analysis of the population response to external stimulation corroborates these observations (Fig. 10e). Finally, analysis of the correlation between excitatory and inhibitory currents to pairs of neurons in our model recapitulates the balanced cancellation of correlations during non-event periods, and its breakdown during events, as reported in the in vitro recordings (compare Figs. 10f to 1f).

The raising of $f_{IE}$ in our biologically realistic plasticity model (Eq. (5)) is qualitatively analogous to a raising of $\theta_{IE}$ in our phenomenological plasticity model (Eq. (3)). Satisfyingly, the population event dynamics from our network of spiking neuron models respond to a raising of $f_{IE}$ similarly to how the phenomenological firing rate model responds to a raising of $\theta_{IE}$ (Fig. 8). Further, by varying the set of plasticity parameters $\{f_{EE}, f_{IE}, d_{EE}, d_{IE}\}$ we can capture the full suite of population event dynamics as observed in the phenomenological model (compare Figs. 7 to S2). In sum, the network of spiking neuron models with realistic short-term synaptic plasticity dynamics population event dynamics with a good agreement to those from our heuristic model. This consistency across models and experiments offers compelling evidence that the mechanisms we describe effectively explain the dynamics observed in cortical circuits.

## Discussion

In this work we studied population dynamics of networks with balanced excitation and inhibition, where excitatory projections exhibit STP. Our models provid a parsimonious explanation of recordings from spontaneously active in vitro slices of auditory cortex[25]. In particular, population events were sporadic and random in time, and depending on STP thresholds population events can have rhythmic or arrhythmic dynamics, all in agreement with analysis of the cortical recordings.

### Synaptic depression and population events in the auditory cortex

Population recordings from auditory cortices of both anesthetized and unanesthetized animals show neuronal activity that transitions

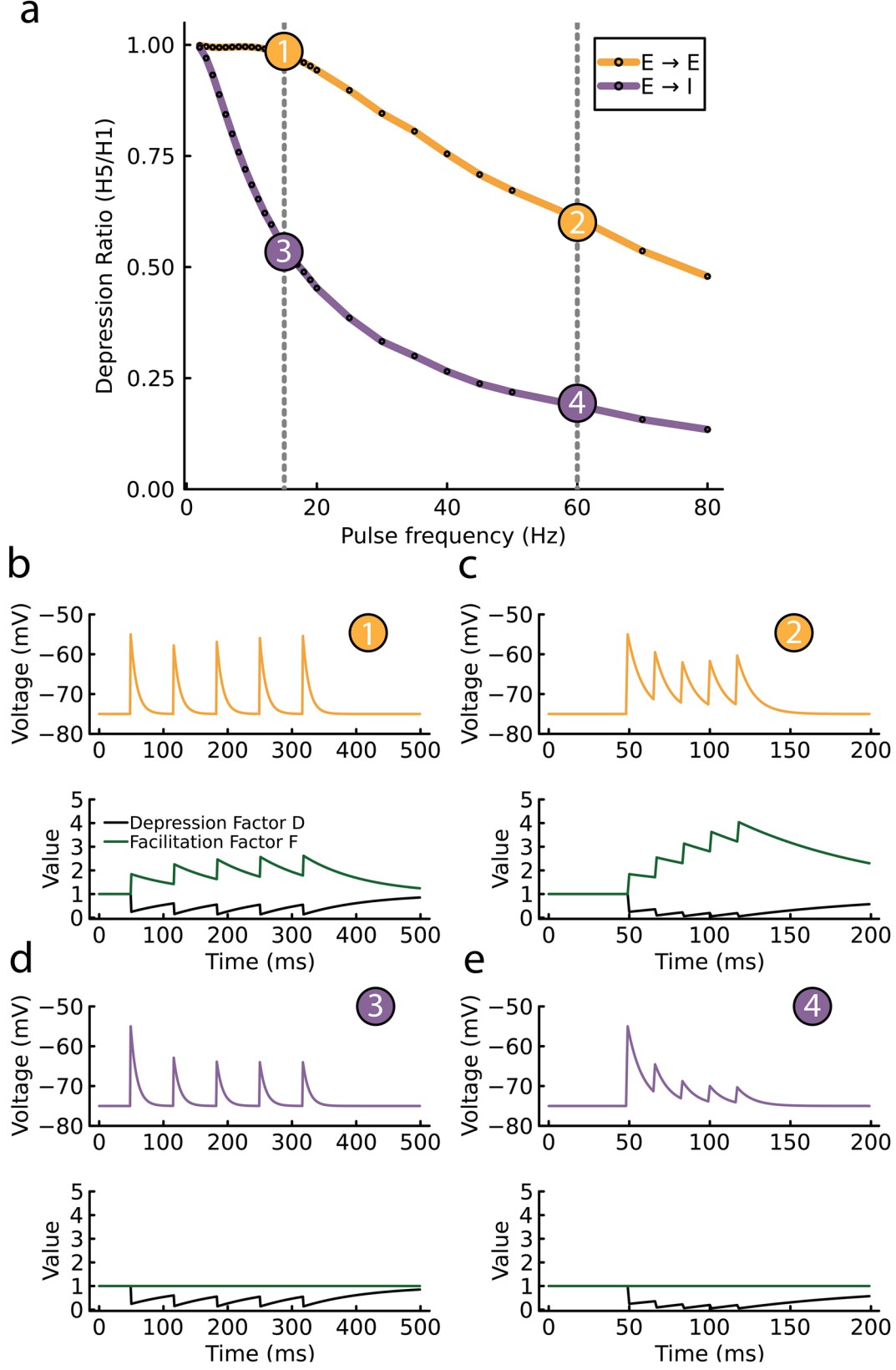

between desynchronized and synchronized states[8,67–69]. In the synchronized state, spontaneous population activity produces periods of brief coordinated spiking activity that is akin to population event behavior[27]. Further, in vivo whole cell recording shows sporadic barrages of excitation that are expected from population event dynamics[26], and are similar to those of the in vitro data presented in

this study (Fig. 1). Unsurprisingly, several past modeling studies have thus proposed core mechanisms for population event activity in the auditory cortex.

While our study focuses upon population events that result from a synaptic weakening of recurrent connections, the auditory cortex nonetheless robustly produces population events as an onset response

**Fig. 9 | Modelling biologically realistic synaptic facilitation and depression.**
**a** Model synaptic response to a five pulse protocol, measuring the response height ratio from the fifth to the first spike (as in Fig. 3b). It compares the $E \to E$ (orange curve) and $E \to I$ (purple curve) synapses, highlighting differences in the recruitment of STP. Both synapse types exhibit a spike-recruited depression ($d = 0.24$); however, $E \to E$ synapses have a spike-recruited facilitation ($f = 0.85$), while $E \to I$ synapses have no facilitation ($f = 0$). **b**–**e** correspond to conditions 1, 2, 3, and 4 in (**a**), illustrating the dynamic responses of $E \to E$ and $E \to I$ synapses under different stimulation frequencies. **b**, **c** set at 15 Hz, show the spike responses for $E \to E$ and $E \to I$ synapses, respectively. In each panel, the upper part displays the membrane potential of the postsynaptic neuron in response to a series of five spikes, while the lower part concurrently charts the dynamics of the Depression Factor (D) and Facilitation Factor (F). **d**, **e** operating at 60 Hz, depict synaptic depression for both types of synapses.

to sustained pure tone inputs[70,71]. It is likely that the physiological mechanisms underlying evoked and spontaneous population events are very related. In tone evoked responses, synaptic depression of $E \to E$ connectivity has long been thought to play a central role, in part from the well known forward masking effect in auditory cortex[72]. Forward masking is the phenomenon by which cortical spiking responses to successive tone inputs show a strong suppression for later tones, and this suppression lasts for hundreds of milliseconds. In vivo whole cell recordings show that the timescale of recurrent inhibition is too fast to explain the suppression phenomenon[70]; however, the recovery time-scale for excitatory synaptic depression is much longer[39,73] and matches the suppression recovery time. Thus motivated, previous recurrent circuit models have captured tone evoked population event dynamics strictly through short-term synaptic depression of excitatory-to-excitatory connections[32,33]. In spontaneous conditions, these models produce sporadic population events with a single excursion in activity, akin to much of our in vitro data set (Fig. 2c) and our network model with $\theta_{EE} < \theta_{IE}$ (Fig. 7, top inset).

Our in vitro recordings also clearly show a subset of population events with rhythmic dynamics. This behavior was not discussed in past experimental studies (including ref. 25), and to our knowledge represents a novel finding. Our data is consistent with in vivo recordings from auditory cortex, where population responses show an augmentation of population responses to successive tones (as opposed to a forward masking) if tone pulsing is restricted to low frequencies[74]. This suggests a network susceptibility for low frequency inputs, of which the spontaneous population event rhythmic dynamic may simply be a reflection. Indeed, the short term facilitation of $E \to E$ synapses in the biophysically realistic model is also consistent with an enhancement of response to low frequency tones. Model networks with strictly excitatory neurons having synaptic depression can show a low frequency (~10 Hz) population oscillation[47,48]. However, the oscillation in these models is sustained, rather than a transition between a low activity regime and a brief, rhythmic population event. In our model, the oscillatory dynamic is mechanistically similar to those of these past studies (Fig. 6); however, strong recurrent inhibition is needed to create an additional stable low activity state (Fig. 5). This complicates how a population event is initiated, since the stable state is not depression stabilized, as was the case in the past models[32,33,48].

We used slow synaptic depression of $E \to I$ synapses to create a saddle point that separates stable low activity from high activity dynamics, and activity fluctuations prompt stochastic crossings of the stable manifold of the saddle point. For this to be the mechanism for population event initiation our model has a strong requirement: the depression of $E \to I$ synapses must be recruited at lower excitatory rates than the depression of $E \to E$ synapses ($\theta_{EE} > \theta_{IE}$ in Fig. 7). Fortunately, there is strong evidence for this from paired recordings in layers II/III and IV of auditory cortex (Fig. 3a and refs. 39,40). There is a precedence of STP being determined by the post-synaptic target[55,75,76], so that a difference between $\theta_{EE}$ and $\theta_{IE}$ is not curious. However, the $\theta_{EE}$ measured in auditory cortex[39] is higher than that generally reported in the neocortex[50,64], making it easier for $\theta_{EE} > \theta_{IE}$ to be satisfied. This difference in $\theta_{EE}$ across datasets may be region dependent, though a likely cause is a shift in STP over the critical development period[39,40], making $\theta_{EE}$ and $\theta_{IE}$ age dependent. Indeed, the recordings we based our measurements of $\theta_{EE}$ and $\theta_{IE}$ upon were recorded in animals that were postnatal day 19–29[39,40], compared to the postnatal day 13–15 used in other studies[50]. In fact, when recordings are restricted to the auditory cortices of younger animals the $\theta_{EE}$ and $\theta_{IE}$ values are lower[39,40], consistent with other datasets. In sum, we view the measurement of $\theta_{EE} > \theta_{IE}$ in auditory cortex as a strong postdiction of our model, since the fact that $\theta_{EE} > \theta_{IE}$ was unremarked upon in past studies[39,40].

### Biophysical sources of variability in synaptic depression
Our model indicates that even small changes in the STD of $E \to I$ synapses can alter whether population events express a slow oscillatory pattern or instead yield a single-peaked transient. In Fig. 8, for instance, two nearby parameter points (III and IV) differ only slightly in $\theta_{IE}$ yet produce oscillatory vs. non-oscillatory events, respectively. A similar outcome arises in our spiking network simulations, where the parameter choices marked by the purple circle (∘) and gold star (⋆) in Fig. 10b yield these same distinct event phenotypes despite modest adjustments to the degree of $E \to I$ depression. Thus, small slice-to-slice fluctuations in synaptic plasticity can explain why some slices show a slow, within-event rhythm while others produce only isolated population spikes.

Numerous factors could naturally give rise to a slice-to-slice variability in STD of excitatory synapses within auditory cortex. Neuromodulators such as acetylcholine (ACh) or norepinephrine (NE) can regulate presynaptic release probabilities and thereby shift synapses from facilitation to depression, with target-specific and layer-dependent effects[77,78]. Additionally, metabolic byproducts like adenosine, which accumulates through ATP breakdown, reduce release probabilities and accentuate depression in many excitatory pathways[79,80]. Experimental procedures also contribute: slicing protocols, incubation conditions, and the degree of tissue viability can all alter extracellular levels of these modulators. For instance, higher endogenous adenosine typically correlates with more pronounced depression, whereas ACh or NE can either amplify or dampen STP depending on receptor subtype and circuit context[77,78]. Moreover, the local microcircuit composition exerts strong influence on net depression, since excitatory synapses onto parvalbumin (PV) interneurons often depress more robustly than those onto somatostatin (SST) interneurons[75,81,82]. Small shifts in PV:SST ratios between slices thus yield meaningful variation in the aggregate STD signature. Furthermore, intrinsic presynaptic heterogeneity–in terms of release probability and vesicle pool size–can cause some synapses to depress more rapidly than others[83]. Overall, the interplay of neuromodulatory signals, metabolic state, and cell-type-specific wiring explains much of the slice-to-slice heterogeneity in $E \to I$ depression, that we predict underlies the diversity of responses reported in experiment (Fig. 2e).

### Balanced networks and population events
The theory of balanced networks is successful at explaining several core features of cortical activity, namely irregular spiking activity[14], large firing rate heterogeneity[18], and asynchronous population activity[8]. More recently, balanced networks have also been shown to produce stimulus tuned responses in randomly wired networks that lack columnar structure[84], they can be critical for selective population codes[85,86], and with structured synaptic wiring, they can produced correlated network activity[6,87,88]. Despite these advances, a linear

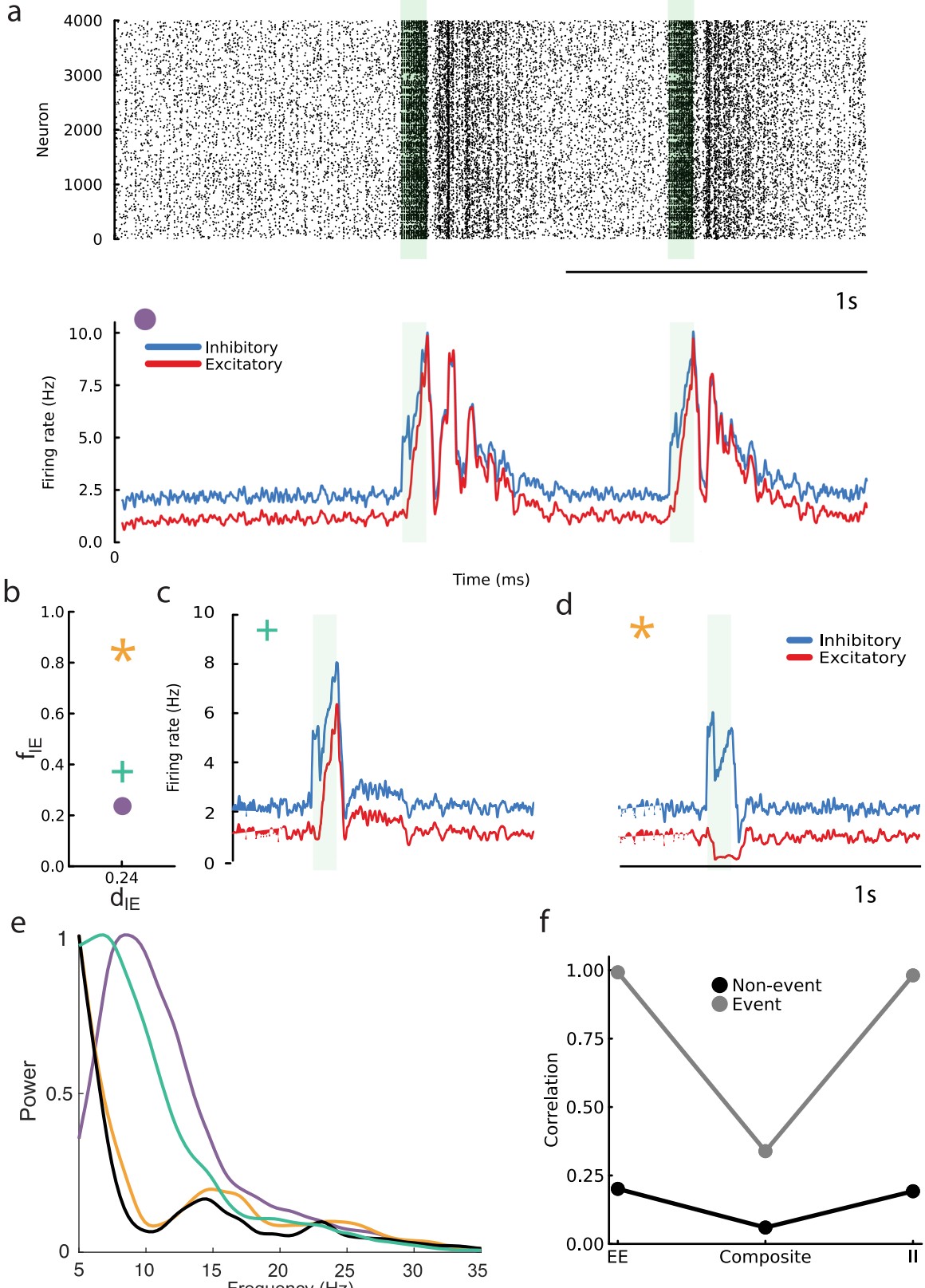

**Fig. 10 | Population events in a network of spiking neuron models with $E \to E$ and $E \to I$ STP. a** *Top:* raster plot featuring oscillatory events. *Bottom:* Firing rate dynamics, excluding the 200 externally stimulated neurons from the excitatory population. The green-shaded region indicates the "kicking" period. **b** The plasticity parameters of the $E \to I$ synapse, with variations only in $f_{IE}$. The symbols in the top left corner of each firing rate plot correspond to different $f_{IE}$ settings used in the simulations: a plus sign (+), a circle (o), and a star (*), each representing distinct plasticity scenarios. **c, d** Firing rate dynamics for the sign (+) and star (*) scenarios. **e** Power spectra for three scenarios differentiated by color-coded symbols, with the black trace indicating a control case without plasticity. **f** Correlations of synaptic inputs from pairs of neurons.

stimulus-response relationship for balanced networks is enforced by the balance condition (see Eq. (1)), and this continues to be a large barrier towards this model framework explaining nonlinear properties of cortical response[89].

Mongillo et al.[38] used STP to impart new nonlinearities to population solutions of balanced networks. In their work, they considered facilitation of the $E \to E$ synapse, which produced a network with stable low and high activity states. This is ideal for neuronal dynamics where integration dynamics are required, such as in models of working memory[90]. By contrast, we use depression of the $E \to I$ pathway to give bistable dynamics. In and of itself, this is not particularly novel since the removal of recurrent inhibition through a depression-mediated weakening of $E \to I$ pathway is similar, in spirit, to a strengthening of the $E \to E$ pathway via facilitation. However, when $E \to E$ depression is also considered, the high state is destabilized and result in an excitable network capable of population event dynamics. This is a population dynamic not previously explored in networks with large recurrent inhibition, instead restricted to networks where inhibition is often ignored[32,48]. Capturing population events in inhibitory stabilized networks offers an important bridge between cortical recordings where recurrent inhibition is a critical aspect of network function[28,34-37,91] with recordings where brief periods of unchecked excitation are clearly apparent[29,30]. Finally, linking population event initiation to $E \to I$ depression allowed $E \to E$ depression to drive rhythmic dynamics within the population spike. Rhythmic population events represent a new population dynamic, not previously discussed in past models, yet it captures a salient feature of our in vitro dataset. In total, our work then broadens the repertoire of nonlinear population dynamics that balanced networks can capture.

In our model, the population events leave the low activity fixed point through endogenous fluctuations modeled as Markov noise in the rate model, while in the network of spiking neuron models, we employ an external brief "kick" to 5% excitatory neurons. These perturbations are illustrative, and likely do not capture the complexity of the slice environment in experiment. Rather, they demonstrate that small, deviations from the low activity state are sufficient to ignite full population events. Experimental and theoretical work shows several plausible sources of such deviations. First, spontaneous miniature synaptic events and asynchronous transmitter release create temporal fluctuations that can perturb a few neurons above threshold, seeding a network-wide burst[92,93]. These fluctuations arise from finite network size effects, stochastic vesicle release, and sporadic firing of highly excitable cells, effectively acting as the "noise" term that displaces the network from rest. Second, slow modulatory processes can drift the synaptic depression parameters across bifurcation boundaries, lowering the threshold for a noise-driven transition. While we lack of clear understanding of the underlying synaptic physiology in our slice experiments, our work explains that very subtle changes in the physiology can be important determinants of the structure of population events (without the depression mechanism, there would be no events, as illustrated by the dynamics in Fig. 10d).

### Phenomenological versus biological modeling of STP

Our modeling effort involved two distinct approaches. Our simplified firing rate model used a heuristic treatment of synaptic depression (Eqs. (1)-(3)). This allowed us to identify that a difference between the thresholds to recruit depression in the $E \to E$ synapse compared to the $E \to I$ synapse is a critical feature needed to capture the rich dynamics reported in experiments. This identification was possible because the low dimensional mean field theory for the rate model allowed a principled analysis of network dynamics. Further, the fact that the depression threshold is a model parameter ($\theta_{\alpha E}$) rather than a measured observable from the network response greatly facilitated model exploration.

The second approach was exploring a large-scale network of spiking neuron models that have biologically realistic STP with the possibility of both facilitation and depression (Eqs. (4) and (5)). This network model captured neuronal fluctuations and neuron-to-neuron covariability characteristic of a low firing rate asynchronous spontaneous state, yet the population transients matched the character of both the data and the phenomenological model. While building a quantitative theory of population events in this realistic network model would be difficult, the qualitative guidance of our phenomenological model allowed a targeted exploration. In particular, the large-scale model makes the prediction that facilitation dynamics in the $E \to E$ synapses, and not in the $E \to I$ synapse, is the key biological featured needed to capture the population event dynamics reported in the data. Such diversity in STP dynamics that is determined by the class of the postsynaptic neuron has been widely reported[39,40,55,75,76].

The motivating dataset for our study showed complex population-wide transients, having either rhythmic or arrhythmic character, while the steady state dynamics were roughly asynchronous across the population (Fig. 2). This is much more complex than the typical network behavior that is modeled, where often only one dynamical regime is considered: i.e., asynchronous population dynamics[7,8], rhythmic synchrony[9], long timescale dynamics[10], and population-wide coordinated behavior[11-13]. Building models that capture such complex steady state and transient behavior requires incorporating multiple aspects of cortical circuit dynamics; in our case, merging balanced excitatory and inhibitory interactions with STP of the synaptic interactions. A more complete analysis of such models will require a mixed modeling approach where sufficiently reduced frameworks provide insight, while biologically realistic models support the plausibility of any proposed mechanism.

Although many of our rate-based simulations produce population events that approach near-saturation firing levels ($r_E \approx 1$), moderate-amplitude population events also sometimes arise, near-threshold regime. Moreover, our framework can be adapted to produce an even broader range of population events amplitudes if so designed. In the large-scale spiking model, the combination of heterogeneous connectivity and more biologically detailed synaptic plasticity often yields smaller-magnitude events below saturation, especially when external drives vary in strength or lengths. This matches the moderate event amplitudes frequently reported in in vivo studies[94], underlining that our overall approach need not be restricted to large-amplitude population events; factors such as noise, connectivity, and facilitation/depression parameters collectively determine the peak rate.

## Methods
### Markov model of network
For the qualitative characteristics we are interested in studying, it is sufficient to study the aggregate population-level behavior of a network rather than the specific microscale instantiation. We model a two population (excitatory, $E$, or inhibitory, $I$) network of binary neurons that can be in either an active (1) or inactive (0) state. We let $n_\alpha(t)$ be the number of active neurons in population $\alpha$ at time $t$, and $N_\alpha$ be the total number of neurons in population $\alpha$. The firing rate (or mean activity) of each population is calculated as the number of neurons in the active state normalized by the total number of neurons in that population.

$$r_\alpha = \frac{n_\alpha}{N_\alpha} \qquad \alpha \in \{E, I\}.$$

Finally, we introduce the probability $P(n, m, t) = \Pr\{n_E(t) = n, n_I(t) = m\}$. We ignore simultaneous transitions and let the probability evolve

according to the birth-death process

$$\frac{dP(n,m,t)}{dt} = T^E_+(n-1,m)P(n-1,m,t) + T^E_-(n+1,m)P(n+1,m,t)$$
$$+ T^I_+(n,m-1)P(n,m-1,t) + T^I_-(n,m+1)P(n,m+1,t)$$
$$- \left[ T^E_-(n,m) + T^E_+(n,m) + T^I_-(n,m) + T^I_+(n,m) \right] P(n,m,t)$$

$$(6)$$

where $T^\alpha_+$ is the transition rate for $n_\alpha$ increasing and $T^\alpha_-$ is the transition rate for $n_\alpha$ decreasing. The boundary conditions are chosen as $P(-1, m, t) = P(n, -1, t) = 0$, with forced upper bound $P(N_E + 1, m, t) = P(n, N_I + 1, t) = 0$. The excitatory transition rates are taken as

$$T^E_+(n,m) = N_E f\left( \sqrt{N_E}\left( j_{EE} p_{EE} \frac{n}{N_E} - j_{EI} \frac{m}{N_I} + I_E \right) \right), \qquad T^E_-(n,m) = n$$

$$(7)$$

and the inhibitory transition rates are taken as

$$T^I_+(n,m) = \frac{N_I}{\tau_I} f\left( \sqrt{N_I}\left( j_{IE} p_{IE} \frac{n}{N_E} - j_{II} \frac{m}{N_I} + I_I \right) \right), \qquad T^I_-(n,m) = \frac{m}{\tau_I}$$

$$(8)$$

where $j_{\alpha\beta}$ is the strength of connection from population $\beta$ to population $\alpha$, $I_\alpha$ is directly applied current to population $\alpha$, $p_{EE}$ and $p_{IE}$ are the plasticity variables, $\tau_I$ is the inhibitory timescale, and $f$ is a sigmoidal transfer function. Following the standard procedure[95,96], this leads to a mean field of the form

$$\dot{r}_E = -r_E + f\left( \sqrt{N_E}(j_{EE}p_{EE}r_E + j_{EI}r_I + I_E) \right)$$
$$\tau_I \dot{r}_I = -r_I + f\left( \sqrt{N_I}(j_{IE}p_{IE}r_E + j_{II}r_I + I_I) \right)$$

$$(9)$$

For concreteness, we take the transfer function $f$ to be

$$f(x) = \frac{1}{1+e^{-x}}.$$

$$(10)$$

We take the excitatory timescale to be unity, and other parameters as $\tau_I = 1, j_{EE} = 2, j_{EI} = 1, j_{IE} = 5, j_{II} = 2, I_E = -0.12, I_I = -0.2$. To assign a time value to the simulations, we interpret 1 time unit of simulation to be equivalent to 10 ms of real time.

Tracking the total number of neurons in the active state for this model is known to reproduce the asynchronous statistics of an Erdős-Renyi coupled network of binary neurons[18,97]. For the qualitative characteristics we are interested in studying, it is sufficient to study the aggregate quantities $n_\alpha$ and $r_\alpha$. We also note that for a dense network, the average number of connections $K$ scales with the system size $N$ such that $K/N \to c, c \in (0,1)$, a constant. A sparse network has $c = 0$, and a fully connected network has $c = 1$. As such, the "system size" parameter $N$ is easily related to the average number of connections $K$.

## Model of a synapse

We consider a phenomenological model of a synapse similar to that seen in ref. 50. Since we are modeling only the mean activity of the network, only the mean synaptic efficacy will affect the governing equations[54]. The dynamics of a synaptic efficacy variable, $p$, are governed by

$$\frac{dp}{dt} = \frac{1-p}{\tau_r} - \frac{a(r_E)p}{\tau_d}$$

$$(11)$$

$$a(r) = \frac{m}{1+e^{-\beta(r-\theta)}}$$

$$(12)$$

The functional form of $a(r)$ was chosen to reproduce the qualitative feature of depression. If we denote the fixed point of Eq. (11) as $\hat{p}$, then when $r \to 0$, $\hat{p} \sim 1$ and as $r \to 1$, $\hat{p}$ approaches a low value (~0.1 in practice). The synaptic parameters were chosen as identical for both kinds of synapse, with the exception that the thresholds $\theta_{EE}, \theta_{IE}$ were different, and allowed to vary in the case of the bifurcation diagrams. We took $\tau_r = 40, \tau_d = 10, m = 2, \beta = 50$. The threshold parameters $\theta_{EE}, \theta_{IE}$ were bounded between 0 and 1. The ordering of synaptic depression shown in Fig. 3 is equivalent to $\theta_{EE} > \theta_{IE}$.

## Simulation of the Markov model

The simulations of the noisy system were done by simulating the Markov process governed by Eqs. (6)–(8), where the plasticity variables were treated as slow variables that evolved deterministically between birth/death events (see ref. 98 for some remarks on dealing with hybrid stochastic systems). This is equivalent to simulating a piecewise deterministic Markov process with constant propensities between birth/death events. Since the typical timescale of the plasticity variables is an order of magnitude larger than the typical timescale of the rate variables, this seems a reasonable approximation. Alternatively, a back of the envelope calculation shows that for $\Delta t \sim \mathcal{O}(1/N)$, a typical change in plasticity is

$$\Delta p \sim 1 - \exp(-\alpha\Delta t) \sim \alpha\Delta t \sim \frac{\tau_d + \tau_r}{\tau_d \tau_r} \cdot \frac{1}{N} \sim \mathcal{O}\left(\frac{1}{10N}\right) \sim \mathcal{O}\left(\frac{1}{4000}\right)$$

for our typical parameter values, suggesting that the plasticity variables, and thus propensities, are effectively constant between birth/death events.

For a fixed value of $r_E$, we can solve Eq. (11) for a closed form solution of $p(r_E)$. Then our algorithm for the simulation is as follows:
1. Choose values for $n_E$, $n_I$ as initial conditions.
2. Initialize $r_E$, $r_I$, $p_{EE}$, and $p_{IE}$, where $r_\alpha = n_\alpha/N_\alpha$ and $p_{\alpha\beta}$ is chosen as the fixed point of Eq. (11) evaluated at $r_E$.
3. Calculate the transition probabilities $T^E_{+/-}$, $T^I_{+/-}$, and construct the Gillespie probability intervals.
4. Find the time of the next ("current") event.
5. Calculate the current values of $p_{EE}$, $p_{IE}$ based on the current values of $r_E$, $r_I$, and $\Delta t$, the time between the current event and the previous event.
6. Decide which type of event happened: (in/de)crementing $n_{E/I}$.
7. Update the rate variables $r_E$, $r_I$.
8. Return to step 3 and repeat until the simulation is complete.

## Bifurcation diagrams

The bifurcation diagrams were computed using XPPAUT[99], simulating the full deterministic system given by Eqs. (9)–(12). A full discussion of computing bifurcations using numerical continuation is a subtle and complicated one beyond the scope of this paper, though XPPAUT is a standard tool for this purpose in applied dynamical systems. Briefly, we highlight the bifurcations of interest to this work. A *saddle-node bifurcation* is when a pair of fixed points, one stable and one unstable, annihilate each other (cf Figs. 5, 7, and 8). An *Andronov−Hopf bifurcation* is the emergence of a limit cycle when a fixed point changes stability via a pair of purely imaginary eigenvalues. A *period-doubling bifurcation* is when the period a limit cycle doubles, and introduces a characteristic winding of the limit cycle over itself. After such a bifurcation, every other peak returns to the same place is phase space (rather than every peak). Throughout the bifurcation analysis, we consider the network model as deterministic. Finite-size stochasticity is introduced separately (Figs. S1 and 4) by treating the network as a birth-death Markov process to generate rare noise-driven excursions predicted by the deterministic phase portrait. We remark that even in our deterministic model, the parameter $N$ appears in the transfer $f$ (see Eqs. (7) and (8)).

## Event detection in data

Please see ref. 25 for a more detailed description of the collection methods. Briefly, we investigate spontaneously active in vitro slices from mouse auditory cortex through paired whole cell patch clamp recordings. Figure 1f was reproduced from ref. 25 by repeating the analysis performed there.

Since recorded membrane potentials and synaptic currents are nonstationary and fluctuate over a wide range of timescales, recorded data was detrended prior to event detection. The detrended data point at time $t$, $\hat{x}(t)$, was calculated as

$$\hat{x}(t) = x(t) - (\{x(t)\}_{t-w/2,\,t+w/2} - \langle x \rangle_L) \tag{13}$$

where $x(t)$ is the raw data point at time $t$, $\langle x \rangle_L$ is the average of the entire recording of length $L$, $\{x(t)\}_{t-w/2,t+w/2}$ is the median of a window of width $w = 3s$ centered at each data point.

To isolate the events, we performed two passes through the data. In the first pass, we isolated potential excursions by looking for periods where the local activity differed from the overall detrended recording average by 5 mV in current-clamp recordings or 0.15 nA in the voltage clamp recordings. These periods were padded by 200 ms before and 500 ms after the threshold crossing. In the second pass, the specific event times were then marked by hand on an event-by-event basis, where the beginning of the event was taken to be 200 ms before the initial upstroke of the excursion (easily identified by eye), and the end of the event was marked by the last point before the data returned to having a roughly flat slope ($>1s$ "flat" region required).

Visual inspection of the events show that a nontrivial fraction (approx. 1/3) display oscillatory behavior during the excursion. To compute the power spectra, we first denoised the data by applying a Savitsky–Golay filter with a first-order polynomial and a window size of 100 data points (a region approximately 10 ms in width) to remove power over 100 Hz. For each event, MATLAB's detrend function was used to remove any linear trend and center the data, to remove excess low power noise. Finally, the power spectrum of each event was computed using MATLAB's pwelch algorithm, which was then normalized to have unit area. The power spectra reported for each slice are the average over all events, then normalized to have maximum value 1, for ease of visualization in the heatmaps. The power spectra for the nonevent periods were computed identically, using each nonevent period of length greater than 100 ms.

For Fig. 2f, there is no natural criteria for choosing the labels "Cell 1" and "Cell 2", since some recordings are for pairs of excitatory currents, some for pairs of inhibitory currents, some recordings have 3 cells simultaneously recorded, etc. As a result, there are a plurality of possible correlation coefficients. To report a single number, we computed the correlation coefficient for 10,000 possible labelings, where the labels were assigned to each cell randomly. The number reported in Fig. 2 ($\rho = 0.67$) is the mean of the distribution of computed correlation coefficients. A representative scatter plot was chosen for panel f.

## Network of spiking neuron models with realistic synaptic plasticity

We consider a network of $N_E$ excitatory neurons and $N_I$ inhibitory neurons. All model parameters are given in Supplementary Table 1 in the supplementary materials.

Individual neuronal dynamics are characterized by a leaky integrate-and-fire rule[100], where the membrane potential for neuron $i$ in population $\alpha \in \{E, I\}$, $V_\alpha^i(t)$, obeys:

$$\frac{dV_\alpha^i}{dt} = \frac{1}{\tau_\alpha}\left(\mu_\alpha^i - V_\alpha^i(t)\right) + I_\alpha^i(t) + \sigma_\alpha \xi_\alpha^i(t) + K_\alpha^i(t).$$

The membrane dynamics are supplemented with a spike-reset rule, where if at time $t$ we have that $V_\alpha^i(t) = V_{\text{th}} \longrightarrow V_\alpha^i(t_+) = V_{\text{re}}$. We label the

time point $t = t_\alpha^{ik}$ if this is the $k$th time neuron $i$ in population $\alpha$ crossed threshold. The collection of these spike times gives the spike train $y_\alpha^i(t) = \sum_k \delta(t - t_\alpha^{ik})$, with $\delta(\cdot)$ being a Dirac "delta" function. After a spike reset, $V_\alpha^i$ is held at $V_{\text{re}}$ for a refractory period $\tau_{\text{ref}}$. The resting potential $\mu_\alpha^i$ is heterogeneous across neurons and assigned values from a uniform distribution:

$$\mu_\alpha^i \sim \text{Uniform}\left(\mu_\alpha^{\text{L}}, \mu_\alpha^{\text{H}}\right).$$

The parameter $\tau_\alpha$ is the membrane time constant for neurons in population $\alpha$. Finally, $\xi_\alpha^i(t)$ is a white noise input with temporal expectations $\langle \xi_\alpha^i(t) \rangle = 0$ and $\langle \xi_\alpha^i(t)\xi_\beta^j(t') \rangle = \delta^{ij}\delta_{\alpha\beta}\delta(t - t')$ with $\delta^{ij}$ and $\delta_{\alpha\beta}$ being Kronecker "delta" functions (i.e., $\delta^{ij} = 1$ if $i = j$ and 0 otherwise; same for $\delta_{\alpha\beta}$). $\sigma_\alpha$ is the intensity of the white noise input for neurons in population $\alpha$. The input $K_\alpha^i(t)$ is described below.

The synaptic current $I_\alpha^i(t)$ for neuron $i$ in population $\alpha$ is given by:

$$I_\alpha^i(t) = \sum_j J_{\alpha\beta}^{ij}(t)S_\beta * y_\beta^j(t) \quad \text{for } \alpha, \beta \in \{E, I\}.$$

The synaptic kernel $S_\beta(t)$ models the postsynaptic response to a spike from a presynaptic neuron in population $\beta$. The kernel $S_\beta(t)$ obeys:

$$S_\beta(t) = \frac{1}{\tau_\beta^d - \tau_\beta^r}\left(e^{-t/\tau_\beta^d} - e^{-t/\tau_\beta^r}\right),$$

where $\tau_\beta^r$ and $\tau_\beta^d$ represent the synaptic rise and decay time constants, respectively; the operation '$*$' denotes convolution, $a*b(t) = \int_{-\infty}^t a(s)b(t - s)ds$. The synaptic amplitude $J_{\alpha\beta}^{ij}(t)$ is given by the decomposition:

$$J_{\alpha\beta}^{ij}(t) = W_{\alpha\beta}^{ij} \bar{J}_{\alpha\beta} D_{\alpha\beta}^j(t) F_{\alpha\beta}^j(t).$$

The adjacency matrix $[W_{\alpha\beta}^{ij}]$ gives the network graph, where element $W_{\alpha\beta}^{ij} = 1$ with probability $\chi_{\alpha\beta}$ and is 0 otherwise. The coefficient $\bar{J}_{\alpha\beta}$ is the baseline synaptic strength from a neuron in population $\beta$ to a neuron in population $\alpha$. Of critical importance are the presynaptic depression, $D_{\alpha\beta}^j(t)$, and facilitation, $F_{\alpha\beta}^j(t)$, factors. These factors depend on the class ($E$ or $I$) and spike activity of the presynaptic neuron $j$, and the class of the postsynaptic neuron $i$.

The depression and facilitation factors for a connection from an excitatory neuron $j$ onto a postsynaptic neuron in population $\alpha$ update at presynaptic spiketime with the rule[64,101]:

$$D_{\alpha E}^j \to D_{\alpha E}^j \cdot d_{\alpha E}, \quad F_{\alpha E}^j \to F_{\alpha E}^j + f_{\alpha E}, \quad d_{\alpha E} \in (0, 1], \quad f_{\alpha E} \in [0, +\infty).$$

Here $d$ and $f$ denote the spike-recruited depression and spike-recruited facilitation, respectively. In between presynaptic spikes, both $D(t)$ and $F(t)$ decay exponentially to 1 with time constants $\tau^D$ and $\tau^F$, respectively:

$$\frac{dD_{\alpha E}^j}{dt} = \frac{1 - D_{\alpha E}^j}{\tau^D}, \quad \frac{dF_{\alpha E}^j}{dt} = \frac{1 - F_{\alpha E}^j}{\tau^F}.$$

Finally, we do not consider STP from inhibitory neurons, so we have that $J_{\alpha I}^{ij} \equiv \bar{J}_{\alpha I}$ (i.e $D_I^j(t) = F_I^j(t) \equiv 1$ for all $j$).

To generate a population event, we provide a temporally brief, external "kick" input $K_E^i(t)$ to a subset ($N_E/20$) of the excitatory neurons with the highest average firing rates. The average firing rate $r_E^i$ for each neuron $i$ is calculated as: $r_E^i = \frac{1}{T}\int_0^T y_E^i(s)ds$ where $T$ is the length of simulations. For subsequent experiments as reported in Fig. 10, we maintain the same network structure (connection weights and resting potentials) and apply external input only to neurons in $\mathcal{S}$.

For these selected neurons, $K_E^i(t) = G_E^i H(t - t_{\text{on}})H(t_{\text{off}} - t)$, with $G_E^i \sim \mathcal{N}(17.1/\tau_E, 5 \times 10^{-2})$, $H(\cdot)$ being a Heaviside function, and $t_{\text{on}}$ and $t_{\text{off}}$ being the start and end times of the kick. Throughout we have

$t_{\text{off}} - t_{\text{on}} = 80$ ms. When calculating the population averaged firing rate for the excitatory neurons (as reported in Fig. 10), the kicked neurons are excluded from the expectation so that we only measure the synaptic response to the population kick.

Network simulations were conducted using Euler–Maruyama integration with a time step of $\Delta t = 0.1$ ms, ensuring precise temporal resolution for dynamic network responses. Over a time step we take $\xi_\alpha^i(t) \sim \mathcal{N}(0,1)/\sqrt{\Delta t}$.

### Reporting summary

Further information on research design is available in the Nature Portfolio Reporting Summary linked to this article.

## Data availability

This study reanalyzes electrophysiological data originally collected and reported in published works by refs. 25,39,57. We do not generate any new data. Because these data were never deposited in a public repository, we do not make them publicly available. However, in keeping with Nature's data sharing policies, we will make every reasonable effort to assist researchers who wish to verify or build upon our findings. Specifically, qualified investigators may contact us directly, and we will coordinate with the data owners to facilitate access under the appropriate data-use agreements and any additional conditions they may require.

## Code availability

The code supporting the findings of this study is available on GitHub. The repository can be accessed at https://github.com/brain-math/events_std.

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

## Acknowledgements
The authors acknowledge the following support: B.D. is supported by the NIH grants U19NS107613-01, R01NS133598, CRCNS-R01EY034723, a Vannevar Bush faculty fellowship (N000141812002), and the Simons Foundation Collaboration on the Global Brain. B.D. benefited from Physics Frontier Center for Living Systems funded by the National Science Foundation (PHY-2317138), and support from the National Institute from Mathematics and Theory in Biology (Simons Foundation award MP-TMPS-00005320 and NSF award DMS-2235451). Y.X. is supported by the NIH training grant R90DA060338. M.G. was supported by a Feodor Lynen Research Fellowship from the Alexander von Humboldt Foundation. B.E. was supported by NSF DMS 1712922. A.R. was supported by NIH 1R01MH129031-01.

## Author contributions
Study conception: B.D., B.E., A.R., and J.D. Data analysis and visualization: J.D. and M.G. Rate-based model: J.D., B.E., and B.D. Spiking neuron model: Y.X. and B.D. Original writing: B.D., J.D., and Y.X. Reviewing and editing: J.D., Y.X., M.G., B.E., A.R., and B.D. Project administration and supervision: B.D. and B.E.

## Competing interests
The authors declare no competing interests.
