## [Transparent Peer Review file · Nature Communications]

Interleaving asynchronous and synchronous activity in balanced cortical networks with short term synaptic depression

Corresponding Author: Professor Brent Doiron

Version 0:

Reviewer comments:

Reviewer #1

(Remarks to the Author)

The goal of this paper is to explain the results of a set of experiments in slice (Graupner et al. 2013). In those experiments, the slice spent most of its time in a low firing rate asynchronous state, but there were occasional brief transitions to a much higher firing rate, more synchronized state.

As pointed out by the authors, such transitions are essentially impossible to explain with a standard network of excitatory and inhibitory cells operating in the strong coupling regime -- where by strong I'm referring to the usual $1/\sqrt{K}$ scaling of the weights, where K is the number of connections, with K large. Additional slow variables are needed, and for that the authors propose synaptic depression from excitatory cells, both to other excitatory cells and to inhibitory cells. There's still a minor problem with this model, as the probability of a transition to a high firing rate state falls off rapidly as K increases. So the authors consider a stochastic model with moderate K .

The paper is solid -- not surprising, given the authors. However, I'm skeptical that their model describes the results of Graupner et al. I looked at that paper, and the firing rates were very low: the reported average firing rate was 0.5 Hz. In general, the average excitatory membrane potential associated with asynchronous activity is

$$\bar{V} = K_E \nu_E \tau V_{\text{psp}}$$

where K_E is the average number of excitatory connections, ν_E is the average firing rate of the excitatory neurons, τ is the membrane time constant, and V_{psp} is the average PSP size. Using $\nu_E = 0.5$, $\tau = 10$ ms, and $V_{\text{psp}} = 1$ mV (the latter is probably on the high side), we have

$$\bar{V} = (K_E/200) \text{ mV.}$$

That's a bit of a problem: if K_E is small enough to generate transitions to a high activity state -- no more than 400 if we are to take Fig. 5 seriously -- then \bar{V} is small; using $K_E = 400$, \bar{V} is only 2 mV. That's small enough to make operation on the unstable branch of the excitatory nullcline, as shown in Fig. 1d, unlikely. In fact, it's small enough to make it in general difficult to understand what's going on. It's possible that their model describes this regime, but I think it's unlikely. I could be convinced by a spiking network model, but not by the birth-death model used in the paper.

In summary: the problem is interesting, the analysis is solid, but it would take simulations with a spiking network to convince me that their model applies to Graupner et al.'s data. Or to any data that I'm aware of.

Reviewer #2

(Remarks to the Author)

This study extends previous models of asynchronous cortical activity interrupted by brief synchronous bursts of activity across the local cortical population (population events = PE), as reported in the auditory cortex in vitro and in vivo.

In addition to their modeling, the authors have re-analyzed the dataset of Graupner and Reyes 2013, in which they performed whole cell recordings in auditory cortical slices. The current authors find systematic differences in dynamics (oscillatory vs. non-oscillatory) between PE's recorded in different slices. They show that a simple modification of previous models of auditory cortical PE's (e.g. those of Refs 6, 27, 28 cited here), namely the addition of synaptic depression to excitatory (E) to inhibitory (I) synapses is sufficient to capture this rich set of PE dynamics. Moreover, they show that in this new model the low-activity asynchronous state is self-consistent and inhibition stabilized (as suggested by a range of studies to be the case in asynchronous phases of cortical state) as opposed to the depression stabilized baseline state of previous models of auditory cortical PE's.

The findings are novel and the study is potentially very valuable to the theoretical neuroscience community and the wider neuroscience community. However, the authors should address a few concerns regarding their modeling choices, in particular the model of synaptic depression employed here and address whether the behavior reported here persists when other more established models of depression are used instead.

Major comments:

- What is the justification for the sigmoidal choice (with a rather sharp threshold) for the function $a(r)$ in Eqs. (2)-(3)? This choice is in contrast to the models in the references they cite (Refs 33, 45, 48 & 66) before introducing Eqs. 2-3, in which this factor depends either linearly on r (e.g. in the absence of short-term facilitation) or at any rate varies much more smoothly with r than their functional form. One reason for my comment is that the phenomenological models developed in those previous publications have been shown to fit very well the behavior of synaptic depression, and moreover the form of $a(r)$ (or equivalent) in the models of Refs. 33, 45, 48 have straightforward biophysical interpretations. The 2nd term in Eq. 2 (in which $a(r)$ appears) represents the neurotransmitter depletion rate due to spiking. This is therefore expected to be proportional to the spiking rate, r , as opposed to sigmoidal. Even if other processes such as Calcium-dependent facilitation of synapses are taken into account, asymptotically (for large r) $a(r)$ should probably grow like $\sim r$, not ~ 1 . Correspondingly, in the models of references 33, 45, 48 & 66, the p -variables behave like $1/r$ for large r and therefore pr is asymptotically constant. By contrast, with the sigmoid behavior of $a(r)$, p becomes asymptotically constant for large r and pr grow linearly with r . (This difference is also visible in Figure 3 of the current manuscript: the empirical depression ratio curves in Fig 3B vary much more smoothly with r , compared to the model's curves in Fig 3C.)

My concern is whether the same bifurcations and rich variety of population events (PE) can be seen if those (arguably more realistic and accurate) models of synaptic depression were used, instead of the approximately-piecewise-constant and threshold-like form for $a(r)$, and whether reasonable variations in the parameters of those (previously published) models could similarly (to variations in the thresholds θ_{EE} and θ_{IE} here) lead to the varieties of PE dynamics (oscillatory vs. non-oscillatory) as in the current model.

I agree that there is (theoretical) value to the current model in that it probably simplifies and elucidates the bifurcation and parameter space analysis. So I'm not suggesting that they do all the analysis using the non-sigmoidal models; just that they establish (e.g. with a few simulations) that the different types of PE can still be reproduced if those previously published models of synaptic depression were used, with reasonable variations in the depression parameters of E->I and E->E synapses.

- The full bifurcation diagram of Figure 8 should be described more thoroughly by providing information about stable states (attractors) in each region. If the Green lines denote super-critical Hopf bifurcations, then why is there no stable limit cycle at point IV of Figure 9? Is this due to the period doubling bifurcations, or another bifurcation? On page 15 the reason is attributed to the crossing of the horizontal saddle-node bifurcation line. (And from figure 9 it appears that this saddle-node bifurcation occurs on the limit cycle.) I suggest the authors add a (supplementary) figure in which they denote what the stable attractors (limit cycle, high-rate fixed point, low-rate fixed point, etc) are at each point/region of the θ_{IE}/θ_{EE} parameter plane. (Related: the purple and green bifurcation lines are partly overlapping in Fig 8. Perhaps they can provide supplementary figures in which these are separately plotted on the θ -plane.)

- This work predicts that differences in the degree of synaptic depression in excitatory synapses (particularly E -> I synapses) are primarily responsible for determining whether population events exhibit oscillatory activity or not. Since we are dealing with a mean field model, the relevant quantity is the average degree of depression across the excitatory synapses in the local cortical network. They should discuss what reasonable variations in biophysical factors can lead to sufficiently large variations in this average quantity (degree of depression) across different slices.

- The model's population events have amplitudes (in terms of average population rate) that in almost all cases (shown in Fig. 8) seem to reach neuronal saturation (corresponding to $r \sim 1$ in their birth-death model). This may be appropriate for the very large amplitude events seen in the in vitro slice dataset of Graupner and Reyes re-analyzed here (see Fig 2A). But the bumps or population events reported in vivo (which the authors discuss in their Intro and Discussion) seem to have smaller amplitudes (see e.g. Refs 20, 21, 59 cited here or Hromadka et al. 2013) suggesting average population rates at the peak of population events (PE) are far from saturation. Can the model capture significant variations in the amplitude of PE's?

Minor comments:

- In the abstract they write: "At the other extreme are models where short term synaptic depression between excitatory neurons can generate the epochs of population-wide activity." But the PE's are the result of strong recurrent excitation in these models, and not the depression. Synaptic depression in E-E is the factor that ends or control the PEs, not the factor responsible for their initiation. So I suggest changing this sentence to "At the other extreme are models in which strong recurrent excitation that is quickly tamed by short term synaptic depression

between excitatory neurons leads to short epochs of population-wide activity.”

- In the first paragraph on page 19 (Discussion) they link the rhythmic dynamics within the PE's reported here to the facilitation of auditory cortical responses to tone trains of low frequency. But if this is indeed the case the response facilitation should peak at a finite (albeit low, $\sim <10\text{Hz}$) frequency of tone pulsing. Is that the case?

- At the end of first section of Discussion they cite Refs. 34-35 reporting stronger depression in younger animals. Can they speculate about predictions of their model in changes in PE dynamics as a function of maturity (post-natal day) or cortical slice preparation?

- In the intro (beginning of page 2) they say that balanced network-like models of asynchronous activity require Gaussian fluctuations “in population activity” and that they are incapable of accounting for the skewed non-gaussian distribution of population activity, and for the latter they cite Ref. 25 (Buzsaki et al. 2014) which reports non-gaussian heavy tailed (approximately log-normal) distributions of average firing rates in cortex. This needs to be corrected.

- Those models are based on a gaussian approximation to the synaptic inputs received by neurons, and not the (quenched) distribution of their rates. In fact balanced networks have been shown (see Roxin et al. J Neuroscience (2011), 31, 16217) to be able to account of the approximately log-normal distribution of average rates across neurons.

- I found the notation $E \rightarrow E$ or $I \rightarrow I$ used on page 4 when referring to Fig. 1 to be a bit confusing. I suggest replacing these (which denote non-directional correlations) with $E-E$ and $I-I$, instead.

- In caption of Fig. 8 they write “vertical bars denote 0.2 (arbitrary units)”. But the plotted quantity are the r 's (activity) of their model which is a dimensionless quantity (fraction of active neurons). Having no units is different than having arbitrary units.

- Typo on line 5 of page 20: “drive rhythmic dynamics within the spike” should be “drive rhythmic dynamics within the population spike”.

- After Eq. 9 on page 21 they write: “Tracking the total number of neurons in the active state for this model is known to reproduce the asynchronous statistics of an Erdős-Renyi network of binary neurons.” They should provide a reference for this claim.

Reviewer #3

(Remarks to the Author)

The authors propose a neural network model of the neocortex that combines two of the design principles widely used in the past: a tight balance between excitation and inhibition and short-term synaptic depression. The phenomenon that the model is designed to explain is the peculiar form of cortical dynamics that is characterized by long epochs of asynchronous activity interrupted by brief epochs of synchronized bursts involving the whole network. Moreover, some of the bursts ('population events') contain oscillations and some not. The authors show that the new model widens the repertoire of different activity states of the network, and allows a better account of the experimental recordings in cortical slices. This will be of interest to people working on cortical network models. I have several issues related to the technical exposition of the model and its analysis. My main problem is that I don't understand why the authors call the equations (1) a 'meanfield', or a limit of an infinite network. This equation is indeed deterministic but has K as a parameter, so it cannot describe an infinite network? When the authors compute the bifurcation diagram of the model, do they choose a particular value for K ? They never mention this in the figure captions. Further to the issue of parameters, the network is analyzed for a particular choice of synaptic strengths, how was this choice made, and how general the results are? Since there is no attempt to relate synaptic strengths to experimental ones, this is a crucial question for the model. Finally, the synaptic depression description in equations (2,3) is unusual because it includes the sigmoid function of the firing rates. This means that for very high rates, depression does not increase as in standard models. The experimental recordings shown in Fig. 3b does not seem to support this feature of the model, in fact the modeling curves look very different from the experimental ones. Concerning the analysis of the model, the authors use bifurcation analysis of the deterministic model and the stochastic version of birth-death Markov process with finite K . The simulation results for the stochastic model are shown in Figure 5, it seems that the population events only emerge for rather small K 's and disappear for biologically realistic values. The authors don't discuss this apparent deficiency of the model. The relation between the simulations the bifurcation analysis is not very clear. The authors should state clearly which of the parameter regions are relevant for the experimentally observed activity regimes. Much of the discussion is devoted to a bistable regime, but this seems to be irrelevant since it does not exhibit brief network events.

Version 1:

Reviewer comments:

Reviewer #1

(Remarks to the Author)

The inclusion of the spiking network definitely made the paper more convincing. However, I believe it still doesn't explain the experiments in Ref. 25 -- one of the main (although not only) motivations for the paper. That's because to produce bursts, they need external drive, for which there is no (obvious) mechanism in slice. I'm not suggesting that they run new simulations or analysis, but they should at least emphasize this in the discussion. My guess is that there's a slow variable driving the system across a bifurcation boundary. For instance, for the bifurcation diagram in Fig. 7, could one get oscillatory bursts if

there was a slow variable driving the system between the dark-green and green regions, and non-oscillatory bursts if there was a slow variable driving the system between the dark-green and purple regions? If so, that would make a very nice story. And just to be clear, I'm emphasizing that they could use a story, but more simulations or analysis are not needed.

Besides that, I have a few comments mainly related to clarity; the authors can ignore me if they think these are bad suggestions.

1. When analyzing the rate model, to what extent is stochasticity associated with the the birth-death model taken into account? My impression is that the conclusions about bifurcations are drawn from the deterministic model, and stochasticity is used to drive transitions. Is that correct? If so, it should be stated. Otherwise, it should be clear what's going on. (And I will freely admit that possibly they said this and I missed it.)

2. The description of what happens in the reduced (3-variable) models was very thorough. But I was missing why all the statements were true. For instance, on page 11,

"Finally, for very large θ_{IE} the network remains bistable, but the saddle point is distant from the low activity state (Fig. 5b)."

Which saddle point? Is this the saddle point in the E-I phase plane? It would be nice if we had some pictures. Maybe nullclines at various values of p_{IE} ? Or maybe they can reduce the system to 2-D by considering fast inhibition, and then make phase-plane plots? The statements made vague sense to me given what I know about E-I network and depression, but it would be nice to get some more detailed intuition.

The same applies to the reduced model with p_{IE} fixed.

3. Is Fig. 7 a cartoon, or quantitatively correct?

4. What's the time constant for recovery of F and D (Eq 5) in the spiking model?

5. p. 20: What does "morally qualitatively" mean? Or is that a typo?

(Remarks on code availability)

Responses to Referees

The authors thank the reviewers for their careful reading of our responses and the revised manuscript. Our rebuttal has clearly been a long road and may perhaps be a record for resubmission tardiness (I checked with the Nature Communications editorial staff and they encouraged our resubmission). Many things have happened since July 2018 when we received your reviews. First, the graduate student that is the lead author has left research academia, and is focused on teaching efforts. Second, a global pandemic occurred. Third, I (Brent Doiron) moved institutions from the University of Pittsburgh to the University of Chicago to build a center for theoretical neuroscience. But perhaps the biggest reason for the lag was the need to develop and analyze a network of spiking neural models with realistic short term plasticity. This was central in addressing the reviewer criticisms, and in truth we felt that the criticisms were quite valid, and we took their advice very seriously. To properly address these criticisms, a new graduate student was brought on to the project last year (who is now co-first author), and he completed these important additions to the work. We also recognize that the reviewers probably have forgotten entirely the details of our study, along with the structure of their criticisms. This will undoubtedly require a novel read of our paper and we apologize for that. In our rebuttal, we have attempted to streamline the presentation of our new modeling section.

We have made changes to the manuscript in response to the reviewers' comments, and we feel our manuscript has significantly improved. Below we list our responses to each detailed comment from the referees and summarize the changes we have made. Reviewer comments are in blue, our responses are in black, and any changes to the manuscript are marked in red.

All three reviewers raised important issues related to the biological plausibility of our network mechanisms. To attempt to respond to these (reasonable) criticisms, we included an entire new section titled 'Population events in networks of spiking neural models with biologically realistic STP'. This section includes presenting our development and analysis of a network of spiking neural models with biologically realistic short term dependent plasticity mechanisms in the *EE* and *IE* synapses. This involved including two new main figures (Figures 9 and 10) and a new Supplemental Figure (Figure S2). In the individual responses to reviewers, we reference this section where appropriate.

We have also included a copy of the manuscript with red labeled changes as a supplementary file for the reviewers. This is to aid the reviewers in evaluation of our changes.

Reviewer 1:

The goal of this paper is to explain the results of a set of experiments in slice (Graupner et al. 2013). In those experiments, the slice spent most of its time in a low firing rate asynchronous state, but there were occasional brief transitions to a much higher firing rate, more synchronized state.

As pointed out by the authors, such transitions are essentially impossible to explain with a standard network of excitatory and inhibitory cells operating in the strong coupling regime – where by strong I'm referring to the usual $1/\sqrt{K}$ scaling of the weights, where K is the number of connections, with K large. Additional slow variables are needed, and for that the authors propose synaptic depression from excitatory cells, both to other excitatory cells and to inhibitory cells. There's still a minor problem with this

model, as the probability of a transition to a high firing rate state falls off rapidly as K increases. So the authors consider a stochastic model with moderate K .

The paper is solid – not surprising, given the authors. However, I’m skeptical that their model describes the results of Graupner et al. I looked at that paper, and the firing rates were very low: the reported average firing rate was 0.5 Hz. In general, the average excitatory membrane potential associated with asynchronous activity is

$$\bar{V} = K_E \nu_E \tau V_{psp}$$

where K_E is the average number of excitatory connections, ν_E is the average firing rate of the excitatory neurons, τ is the membrane time constant, and V_{psp} is the average PSP size. Using $\nu_E = 0.5$, $\tau = 10$ ms, and $V_{psp} = 1$ mV (the latter is probably on the high side), we have

$$\bar{V} = K_E / 200 \text{ mV}.$$

That’s a bit of a problem: if K_E is small enough to generate transitions to a high activity state – no more than 400 if we are to take Fig. 5 seriously – then \bar{V} is small; using $K_E = 400$, \bar{V} is only 2 mV. That’s small enough to make operation on the unstable branch of the excitatory nullcline, as shown in Fig. 1d, unlikely. In fact, it’s small enough to make it in general difficult to understand what’s going on. It’s possible that their model describes this regime, but I think it’s unlikely. I could be convinced by a spiking network model, but not by the birth-death model used in the paper.

In summary: the problem is interesting, the analysis is solid, but it would take simulations with a spiking network to convince me that their model applies to Graupner et al.’s data. Or to any data that I’m aware of.

We thanks the reviewer for their very careful analysis/critique of the biological plausibility of our model. We feel that we have partially addressed their concerns in our new section network of spiking neural models entitled “Population events in networks of spiking neural models with biologically realistic STP”.

In this section, we develop and analyze a large-scale network of spiking neurons with leaky integrate-and-fire neuronal dynamics and short-term synaptic plasticity (depression and facilitation) following Varela et al. (1997). This more realistic model confirms the main findings of our simpler rate-based approach:

- **Low-rate asynchronous baseline:** Even when the average excitatory firing rates are low (in the network, mean excitatory rate of 1.13 Hz), the network remains in a stable low-activity state, balanced by strong inhibition.
- **Sporadic population events:** Small perturbations (i.e. random “kicks”) can trigger transient runaway activity in the network of spiking neuron models, causing brief, high firing rate population-wide event. Specifically, we give a depolarizing input to a small fraction of neurons (200 of 4000; so 5%) and observe large scale population events in the remaining 3800 neurons. We admit that this is not an internally generated finite size effect as in the Markov network. However, it is unclear how such events are generated in the real *in vitro* network, and a partial synchronized input to a small subset of neurons is certainly plausible. We do show that without short term depression/facilitation that such weak perturbation do not yield large scale population events.

In our resubmitted manuscript we have shifted the finite size analysis of the Markov model to the supplementary section. This was done, in part, to address the reviewers legitimate back of the envelop analysis of internally generated events, and rather re-focus the discussion on how short term synaptic plasticity shapes the dynamics of population events.

- **Oscillatory vs. non-oscillatory population events:** By adjusting the facilitation parameters in $E \rightarrow E$ and $E \rightarrow I$ synapses, our network of spiking neuron models reproduces both single-peak population events and population events with slow ($\sim 2-10$ Hz) rhythmicity, mirroring the heterogeneous event structure seen in slice data.

Reviewer 2:

The findings are novel and the study is potentially very valuable to the theoretical neuroscience community and the wider neuroscience community. However, the authors should address a few concerns regarding their modeling choices, in particular the model of synaptic depression employed here and address whether the behavior reported here persists when other more established models of depression are used instead.

Major comments:

1. What is the justification for the sigmoidal choice (with a rather sharp threshold) for the function $a(r)$ in Eqs. (2)-(3)? This choice is in contrast to the models in the references they cite (Refs 33, 45, 48 & 66) before introducing Eqs. 2-3, in which this factor depends either linearly on r (e.g. in the absence of short-term facilitation) or at any rate varies much more smoothly with r than their functional form. One reason for my comment is that the phenomenological models developed in those previous publications have been shown to fit very well the behavior of synaptic depression, and moreover the form of $a(r)$ (or equivalent) in the models of Refs. 33, 45, 48 have straightforward biophysical interpretations. The 2nd term in Eq. 2 (in which $a(r)$ appears) represents the neurotransmitter depletion rate due to spiking. This is therefore expected to be proportional to the spiking rate, r , as opposed to sigmoidal. Even if other processes such as Calcium-dependent facilitation of synapses are taken into account, asymptotically (for large r) $a(r)$ should probably grow like r , not 1 . Correspondingly, in the models of references 33, 45, 48 & 66, the p -variables behave like $1/r$ for large r and therefore pr is asymptotically constant. By contrast, with the sigmoid behavior of $a(r)$, p becomes asymptotically constant for large r and pr grow linearly with r . (This difference is also visible in Figure 3 of the current manuscript: the empirical depression ratio curves in Fig 3B vary much more smoothly with r , compared to the model's curves in Fig 3C.).

My concern is whether the same bifurcations and rich variety of population events (PE) can be seen if those (arguably more realistic and accurate) models of synaptic depression were used, instead of the approximately-piecewise-constant and threshold-like form for $a(r)$, and whether reasonable variations in the parameters of those (previously published) models could similarly (to variations in the thresholds θ_{EE} and θ_{IE} here) lead to the varieties of PE dynamics (oscillatory vs. non-oscillatory) as in the current model. I agree that there is (theoretical) value to the current

model in that it probably simplifies and elucidates the bifurcation and parameter space analysis. So I'm not suggesting that they do all the analysis using the non-sigmoidal models; just that they establish (e.g. with a few simulations) that the different types of PE can still be reproduced if those previously published models of synaptic depression were used, with reasonable variations in the depression parameters of $E \rightarrow I$ and $E \rightarrow E$ synapses.

We thank the reviewer for questioning whether the rich variety of population events (PEs) in our model depends critically on the threshold-like, piecewise-constant form of $a(r)$, and whether a more realistic synaptic depression model could produce similar oscillatory and non-oscillatory PEs. To address this concern, we implemented a network of spiking neuron models in which synaptic depression depends more gradually on the firing rate (i.e. it is not hard-thresholded). Following Varela et al. (1997), each synapse is governed by depression $D(t)$ and facilitation $F(t)$ factors, each recruited proportionally to presynaptic spikes and recovering exponentially between them. Importantly, we chose parameters so that excitatory–inhibitory ($E \rightarrow I$) synapses still depress at lower rates than excitatory–excitatory ($E \rightarrow E$) synapses, mirroring the key distinction used in our rate model.

Our new simulations are presented in the section “Population events in networks of spiking neural models with biologically realistic STP” (with new Figures 9 and 10 and Supplementary Figure S2) demonstrate not only that both oscillatory and single-peaked PEs can emerge under biophysically realistic synaptic plasticity, but also that all major dynamical regimes predicted by the rate-based model (low state, excitable regime, bistability, periodic dynamics, oscillatory events, and a saturated state) can be recovered by systematically tuning the relevant $E \rightarrow E$ and $E \rightarrow I$ parameters in the spiking model with biophysically realistic STP (Figure S2). Thus, the threshold-like shape of $a(r)$ in Eqs. (2)–(3) is not strictly necessary for capturing these rich dynamics; rather, the crucial requirement is that $E \rightarrow I$ synapses become strongly depressed at lower presynaptic rates than $E \rightarrow E$ synapses. In keeping with our theoretical analysis, we have replicated each of the six dynamical outcomes (see Figure 7 for rate-based bifurcations and Figures 10 and S2 for the corresponding spiking simulations), confirming that both oscillatory and non-oscillatory PEs—and all other states—arise robustly across a wide variety of STP functional forms, provided $E \rightarrow I$ and $E \rightarrow E$ differ suitably in their depression thresholds.

2. The full bifurcation diagram of Figure 8 should be described more thoroughly by providing information about stable states (attractors) in each region. If the Green lines denote super-critical Hopf bifurcations, then why is there no stable limit cycle at point IV of Figure 9? Is this due to the period doubling bifurcations, or another bifurcation? On page 15 the reason is attributed to the crossing of the horizontal saddle-node bifurcation line. (And from figure 9 it appears that this saddle-node bifurcation occurs on the limit cycle.) I suggest the authors add a (supplementary) figure in which they denote what the stable attractors (limit cycle, high-rate fixed point, low-rate fixed point, etc) are at each point/region of the θ_{IE}/θ_{EE} parameter plane. (Related: the purple and green bifurcation lines are partly overlapping in Fig 8. Perhaps they can provide supplementary figures in which these are separately plotted on the theta-plane.)

We agree that a more detailed classification of the attractors is required. In the revised Figure. 7 (formerly Figure. 8), we have added the tuple

$$(\# \text{stable eq, } \# \text{unstable eq, } \# \text{stable lc})$$

for representative parameter points to quantify the attractor configurations in each region. While our colored zones reflect the main bifurcation boundaries, they do not provide a fully comprehensive or rigorously complete classification.

For the specific question here, crossing the “saddle-node” boundary from point III to point II in Figure 8 (formally Figure 9) can cause a stable equilibrium and an unstable equilibrium to collide and give way to a stable limit cycle likely through a homoclinic bifurcation. Consequently, we observe a transition from “(1 stable eq + 2 unstable eq)” to “(1 stable lc + 1 unstable eq).” Moreover, moving vertically from the “yellow” region to point IV crosses a more complicated domain where additional unshown bifurcations may occur. These intricate transitions are not our primary focus. Overall, our simplified bifurcation plots are meant to bridge the deterministic (noiseless) and the finite- K (noisy) perspectives, rather than providing a meticulous classification of every bifurcation.

3. This work predicts that differences in the degree of synaptic depression in excitatory synapses (particularly E \rightarrow I synapses) are primarily responsible for determining whether population events exhibit oscillatory activity or not. Since we are dealing with a mean field model, the relevant quantity is the average degree of depression across the excitatory synapses in the local cortical network. They should discuss what reasonable variations in biophysical factors can lead to sufficiently large variations in this average quantity (degree of depression) across different slices.

We thank the reviewer for suggesting that we clearly point out that multiple biophysical factors can lead to large slice-to-slice variations in this parameter. In response, we have added a new subsection to the Discussion, titled “Biophysical Sources of Variability in Synaptic Depression,” to elaborate on these factors in more detail.

Our work predicts that differences in the average degree of synaptic depression across excitatory synapses—particularly in E \rightarrow I connections—critically determine whether PEs exhibit oscillatory activity. While our mean-field model treats this average depression as a parameter, it is important to note that the slice-to-slice variation in this parameter is small rather than large. In our Revised Figure 8, we highlight two points in parameter space (labeled III and IV) that generate, respectively, oscillatory and non-oscillatory population events. These points lie very close together in the $(\theta_{EE}, \theta_{IE})$ plane, illustrating how a small shift in parameters can switch the event’s character from oscillatory to single-peaked, consistent with the variety seen in experiments. Similarly, in Revised Figure 10(b), the purple circle (o) and the gold star (*) mark two parameter settings in our network of spiking neuron models that reproduce the same experimentally relevant behaviors with a small shift.

In particular, we believe that differences in the neuromodulatory and metabolic state of each slice may play a primary role, while local circuit composition and intrinsic presynaptic heterogeneity may also contribute. Although our model abstracts the average degree of depression as a parameter, this abstraction is well supported by experimental findings (e.g., Kuczewski et al., 2005; Salgado et al., 2011; Dunwiddie and Masino, 2001; Dobrunz and Stevens, 1997; Reyes et al., 1998), which

demonstrate that reasonable variations in these biophysical factors can lead to significant differences in synaptic depression across slices.

The neuromodulatory milieu (e.g., acetylcholine, norepinephrine, adenosine) has been shown to dynamically regulate short-term synaptic plasticity in auditory cortex. For instance, activation of muscarinic acetylcholine receptors can differentially modulate synaptic transmission, enhancing facilitation in some pathways while suppressing it in others (Kuczewski et al., 2005). Similarly, variations in endogenous norepinephrine levels have been reported to alter the balance of short-term dynamics in cortical slices (Salgado et al., 2011). Moreover, adenosine accumulation in slices—due to ATP breakdown—can decrease presynaptic release probability and thereby modify the extent of depression (Dunwiddie and Masino, 2001). These neuromodulatory and metabolic factors are sensitive to the slicing procedure and *in vitro* conditions, leading to marked differences in the effective synaptic depression measured across slices.

In addition to neuromodulation, variability in the local circuit makeup also contributes to differences in short-term plasticity. In layer IV of auditory cortex, the relative proportions of interneuron subtypes (e.g., PV versus SST/VIP interneurons) shape the overall E→I dynamics. Experimental studies have demonstrated that pyramidal neurons form strongly depressing synapses onto PV interneurons but facilitating synapses onto SST interneurons (Reyes et al., 1998; Keijser and Sprekeler, 2023). Furthermore, intrinsic presynaptic heterogeneity—in terms of release probability and vesicle pool size—can cause some synapses to depress more rapidly than others (Dobrunz and Stevens, 1997). Together, these factors lead to variations in the average degree of depression across slices.

4. The model’s population events have amplitudes (in terms of average population rate) that in almost all cases (shown in Fig. 8) seem to reach neuronal saturation (corresponding to $r = 1$ in their birth-death model). This may be appropriate for the very large amplitude events seen in the *in vitro* slice dataset of Graupner and Reyes re-analyzed here (see Fig 2A). But the bumps or population events reported *in vivo* (which the authors discuss in their Intro and Discussion) seem to have smaller amplitudes (see e.g. Refs 20, 21, 59 cited here or Hromadka et al. 2013) suggesting average population rates at the peak of population events (PE) are far from saturation. Can the model capture significant variations in the amplitude of PE’s?

In our current rate-based model, population events (PEs) indeed tend to have large amplitudes; however, moderate-amplitude bursts can also arise. For instance, in Figure R1 we highlight two PEs taken from the same simulation (and hence with identical parameters): one peaks at $r \approx 0.91$, while another remains nearer $r \approx 0.30$. Achieving still greater variability in event magnitudes within the rate-based framework can be challenging due to the relatively rigid, sigmoid-like short-term plasticity functions. By contrast, in our network simulations, the combination of heterogeneous connectivity and more detailed synaptic dynamics naturally yields broader amplitude distributions for PEs, in line with the reviewer’s suggestion.

We have added a brief paragraph at the end of the ‘Phenomenological versus biological modeling of STP’ subsection in the Discussion clarifying that our framework indeed allows population events of varying amplitude, including moderate peak firing rates that are consistent with *in vivo* data.

Minor comments:

Figure R1: Variations of amplitudes in the rate model simulation.

1. In the abstract they write: "At the other extreme are models where short term synaptic depression between excitatory neurons can generate the epochs of population-wide activity." But the PE's are the result of strong recurrent excitation in these models, and not the depression. Synaptic depression in E-E is the factor that ends or control the PEs, not the factor responsible for their initiation. So I suggest changing this sentence to "At the other extreme are models in which strong recurrent excitation that is quickly tamed by short term synaptic depression between excitatory neurons leads to short epochs of population-wide activity."

We have amended the abstract as suggested

2. In the first paragraph on page 19 (Discussion) they link the rhythmic dynamics within the PE's reported here to the facilitation of auditory cortical responses to tone trains of low frequency. But if this is indeed the case the response facilitation should peak at a finite (albeit low, $<10\text{Hz}$) frequency of tone pulsing. Is that the case?

The reviewer raises an interesting prediction. Indeed, to replicate oscillatory PEs our network of spiking neuron models required a synaptic facilitation term for $E \rightarrow E$ synapses that was operative at low input/pulse frequencies. We do not know if this is truly the case since in experiments pulse frequencies below 10 Hz were not used (Figure 3b), however our simulations of the network of spiking neuro models does indeed predict this. We have added a sentence indicating this to the identified section.

3. At the end of first section of Discussion they cite Refs. 34-35 reporting stronger depression in younger animals. Can they speculate about predictions of their model in changes in PE dynamics as a function of maturity (post-natal day) or cortical slice preparation?

We appreciate the suggestion to extend our model's predictions to changes in PEs as cortical circuits mature. While previous work (Refs. 34–35) indicates stronger depression in younger animals, it remains unclear how this heightened plasticity differentially affects $E \rightarrow E$ versus $E \rightarrow I$ synapses. Our model hinges critically on the relative difference in depression thresholds for those synaptic pathways. Without a precise quantification of how each pathway's short-term plasticity scales with postnatal age, it is challenging to predict definitively whether PEs would become more frequent, more oscillatory, or even disappear with stronger depression.

4. In the intro (beginning of page 2) they say that balanced network-like models of asynchronous activity require Gaussian fluctuations “in population activity” and that they are incapable of accounting for the skewed non-gaussian distribution of population activity, and for the latter they cite Ref. 25 (Buzsaki et al. 2014) which reports non-gaussian heavy tailed (approximately log-normal) distributions of average firing rates in cortex. This needs to be corrected. - Those models are based on a gaussian approximation to the synaptic inputs received by neurons, and not the (quenched) distribution of their rates. In fact balanced networks have been shown (see Roxin et al. J Neuroscience (2011), 31, 16217) to be able to account of the approximately log-normal distribution of average rates across neurons.

We thank the reviewer for catching this error. Indeed, the Roxin et al. paper clearly shows that Gaussian input fluctuations can give rise to skewed (i.e log-normal like) distributions of firing rates when an expansive nonlinear neuronal transfer function is considered. We have removed the incorrect citation.

5. I found the notation $E \rightarrow E$ or $I \rightarrow I$ used on page 4 when referring to Fig. 1 to be a bit confusing. I suggest replacing these (which denote non-directional correlations) with $E-E$ and $I-I$, instead.

We have updated it.

6. In caption of Fig. 8 they write "vertical bars denote 0.2 (arbitrary units)". But the plotted quantity are the r 's (activity) of their model which is a dimensionless quantity (fraction of active neurons). Having no units is different than having arbitrary units.

We have corrected the caption to read: "vertical bars denote increments of 0.2 in the fraction of active neurons (dimensionless)." This clarifies that r has no physical units but represents a normalized activity level.

7. Typo on line 5 of page 20: "drive rhythmic dynamics within the spike" should be "drive rhythmic dynamics within the population spike".

This has been corrected as suggested.

8. After Eq. 9 on page 21 they write: ‘Tracking the total number of neurons in the active state for this model is known to reproduce the asynchronous statistics of an Erdos-Renyi network of binary neurons.’ They should provide a reference for this claim.

Thank you for the correction. We have now cited vanVreeswijk and Sompolinsky (1998) and Ginzburg and Sompolinsky (1994).

Referee 3:

I have several issues related to the technical exposition of the model and its analysis.

1. My main problem is that I don’t understand why the authors call the equations (1) a ‘mean-field’, or a limit of an infinite network. This equation is indeed deterministic but has K as a parameter, so it cannot describe an infinite network? When the authors compute the bifurcation diagram of the model, do they choose a particular value for K ? They never mention this in the figure captions.

Equation (1) is derived by considering a large but finite population of binary neurons and then applying a mean-field (law-of-large-numbers) approximation for the fraction of neurons that are “active.” Specifically, each neuron in population $\alpha \in \{E, I\}$ receives input from $\mathcal{O}(K)$ other neurons, and the strength of each connection is scaled as $j_{\alpha\beta}/\sqrt{K}$. As $K \rightarrow \infty$, the aggregate (summed) input to each neuron remains of order 1 (not diverging) and fluctuations vanish in this limit.

Under these conditions, the population firing rates obey a set of deterministic ODEs — the “mean-field limit” — given by Eq. (1). This is a standard procedure in balanced-network theory: the infinite-size limit washes out the stochastic fluctuations at the population level, leaving a deterministic description of the average activity.

Even though we retain the symbol K in Eq. (1), what really matters is that the synaptic coupling scales as $1/\sqrt{K}$. For a truly infinite network ($K \rightarrow \infty$), those terms in Eq. (1) converge to finite values, and the model becomes purely deterministic. We keep K explicit in the notation mainly to show the scaling regime: as $K \rightarrow \infty$, the per-connection strength goes like $1/\sqrt{K}$. That scaling ensures a meaningful balanced-state limit. In practice, once K is large, the population behavior is well-described by the deterministic ODEs.

In our work, we do consider finite K explicitly via a birth–death Markov process (hence allowing random, noise-driven population events). Yet to analyze the deterministic flow (e.g. bifurcation structure), we let $K \rightarrow \infty$ and thereby obtain Eq. (1). That is why Eq. (1) is often called the “mean-field limit” or “infinite-network limit,” even though we keep a K -dependent normalization to track how the system approaches that limit.

However, the reviewer raises an issue that could lead to confusion. As such we do not discuss Eq. (1) in terms of $K \rightarrow \infty$. Furthermore, regarding the choice of K for bifurcation diagrams, in **Figures 5 and 6** we use $K = 400$ for both the bifurcation analysis and the Markov simulations, so that the deterministic and finite- K cases match. However, in **Figure 7**, we maintain $K = 400$ for the Markov simulations but set $K = 1000$ for the bifurcation calculations. This K choice highlights

the parameter regions more clearly in the deterministic limit. We now indicate these distinctions in the captions of the relevant figures. In addition, we emphasize that our bifurcation plots are intended to connect the deterministic (noiseless) and finite- K (noisy) perspectives, rather than to provide an exhaustive classification of every possible bifurcation.

2. Further to the issue of parameters, the network is analyzed for a particular choice of synaptic strengths. How was this choice made, and how general are the results? Since there is no attempt to relate synaptic strengths to experimental ones, this is a crucial question for the model.

We thank the reviewer for highlighting this point. In our firing-rate/birth–death network model (Eqs. (1)–(3)), we selected synaptic strengths $\{j_{\alpha\beta}\}$ to ensure inhibition-stabilized (IS) dynamics that match, in broad terms, *in vitro* conditions where recurrent inhibition strongly counters excitation. Specifically, we take j_{EE} to be sufficiently large so that in the absence of dynamic inhibition that network is unstable. This then requires large enough j_{EI} and j_{IE} so that the network’s asynchronous state is primarily maintained by inhibition stabilization. Beyond requiring a robust excitatory–inhibitory balance, we did not rely on atypical values for synaptic coupling.

Hence, the qualitative phenomena—random population events and their (a)rhythmic character—do not depend on a unique or narrowly tuned set of coupling values. We chose nominal $\{j_{\alpha\beta}\}$ that keep the model tractable for mathematical analysis while preserving the core behaviors. As the reviewer notes, we have not matched these parameter choices to exact experimental measurements (e.g. in pA or mV); rather, we adopted a phenomenological scale where the network’s baseline is inhibition-stabilized, and where excitatory short-term depression can release that baseline for brief coordinated events.

We fully agree that demonstrating generality and linking to experimental values is important. Therefore, in the revised manuscript we added a new section detailing a spiking network with biologically realistic short-term plasticity at EE and IE synapses. We incorporate experimentally motivated parameters for depression and facilitation (d, f at $E \rightarrow E$ vs. $E \rightarrow I$) and choose baseline synaptic strengths that keep the network in an inhibition-balanced, asynchronous regime.

Although our rate-based model uses dimensionless synaptic strengths for simplicity, its core findings—rare population events with oscillatory vs. non-oscillatory structure—remain robust across a broad range of coupling values. Our spiking network simulations confirm that these phenomena do not rely on tight parameter tuning and can be tied to known STP parameters from auditory cortical slice data.

3. Finally, the synaptic depression description in equations (2,3) is unusual because it includes the sigmoid function of the firing rates. This means that for very high rates, depression does not increase as in standard models. The experimental recordings shown in Fig. 3b do not seem to support this feature of the model; in fact, the modeling curves look very different from the experimental ones.

We fully acknowledge that the logistic shape of the depression term is not a point-for-point match to the experimental traces in Fig. 3b, nor was it intended to capture every biophysical detail of short-term plasticity. Rather, this “phenomenological” threshold-like switch enables a low-dimensional description that is mathematically tractable and that illuminates how population events emerge and vary (oscillatory vs. non-oscillatory). Crucially, our analytical treatment shows that a higher effective “depression threshold” for $E \rightarrow E$ compared to $E \rightarrow I$ is essential for reproducing experimental observations of brief yet strong population events.

To verify that our conclusions do not hinge on the specific logistic form, we introduced a more realistic short-term plasticity mechanism in the new spiking network section, “Population events in networks of spiking neural models with biologically realistic STP”. Here we follow Varela et al. (1997), where each synapse’s weight $J_{ij}(t)$ is the product of a baseline \bar{J}_{ij} and two dynamical factors $D(t)$ (depression) and $F(t)$ (facilitation), each updated by presynaptic spikes and recovering exponentially between spikes. By assigning parameter sets that cause $E \rightarrow I$ synapses to depress at lower rates than $E \rightarrow E$ synapses, we again find large-scale population bursts that can be either single-peaked or internally oscillatory. Thus, although Eqs. (2)–(3) do not match every detail of Fig. 3b, the key mechanism—namely that lower depression threshold on $E \rightarrow I$ unbinds strong excitation—remains robust in a fully biophysical STP framework.

4. Concerning the analysis of the model, the authors use bifurcation analysis of the deterministic model and the stochastic version of birth–death Markov process with finite K . The simulations results for the stochastic model are shown in Figure 5; it seems that the population events only emerge for rather small K ’s and disappear for biologically realistic values. The authors don’t discuss this apparent deficiency of the model.

We appreciate the reviewer’s comment regarding the birth–death Markov simulations, which indeed show fewer population events as K (the effective network size) increases. This reflects a fundamental property of fluctuation-driven events in finite-size systems: for larger K , stochastic variability is smaller, making it less likely for the network to escape the low-activity state. In the strict $K \rightarrow \infty$ limit, such “noise-driven” events vanish altogether.

However, the parameter K should not be interpreted as the literal number of neurons in the slice. Rather, it captures an effective scale of synaptic coupling and intrinsic variability. Thus, even if a real slice has thousands of cells, the effective K may be moderate, depending on factors like baseline synaptic strength, the number of connections per neuron, the distribution of synaptic strength (see Ahmadian and Miller, Neuron, 2021). Our simulations show that, for such moderate K , the birth–death system yields exponentially distributed inter-event intervals resembling experiments.

In our new network of spiking neuron models (with 4000 excitatory and 1000 inhibitory neurons), we see that a small external “kick” to a fraction of excitatory cells can reliably trigger a global population event — even though the network is large. Slight parameter changes toggle whether these events contain oscillations or not, matching our core predictions. Hence, while purely noise-driven population events vanish asymptotically for very large K , modest effective network sizes or small perturbations suffice to create the sporadic bursts seen *in vitro*.

To overstate the finite-size, internally induced induced population events we have placed the analysis

of the K dependence of the Markov Model to the supplemental section (new section "Finite Size Scaling of Rare Population Events" and Figure S1).

5. The relation between the simulations and the bifurcation analysis is not very clear. The authors should state clearly which of the parameter regions are relevant for the experimentally observed activity regimes. Much of the discussion is devoted to a bistable regime, but this seems irrelevant since it does not exhibit brief network events.

We thank the reviewer for prompting us to clarify which parts of the bifurcation diagram relate directly to experimental observations. In our revised Fig. 8, we highlight specific parameter points (labeled III and IV) that exhibit, respectively, oscillatory and non-oscillatory population events. These two points lie close to each other in $(\theta_{EE}, \theta_{IE})$ -space, consistent with the idea that minor parameter changes can produce qualitatively different event shapes — precisely as observed in slice experiments.

Likewise, in our simulations of the network of spiking neuron models (revised Fig. 10b), we show that small shifts in short-term plasticity parameters can toggle between oscillatory events (purple circle) and single-peaked events (gold star). Thus, our main focus is on the parameter regimes that yield excitable or oscillatory transients in the model, rather than pure bistability. We do include a discussion of bistable solutions for completeness, but, as the reviewer notes, those do not explain the brief runaways that motivate our study.

Responses to Reviewers

The authors thank the reviewers for their careful reading of our responses and the revised manuscript.

We have made changes to the manuscript in response to the reviewers' comments, and we feel our manuscript has significantly improved. Reviewer comments are in blue, our responses are in black.

Reviewer 1:

The inclusion of the spiking network definitely made the paper more convincing. However, I believe it still doesn't explain the experiments in Ref. 25 – one of the main (although not only) motivations for the paper. That's because to produce bursts, they need external drive, for which there is no (obvious) mechanism in slice. I'm not suggesting that they run new simulations or analysis, but they should at least emphasize this in the discussion.

My guess is that there's a slow variable driving the system across a bifurcation boundary. For instance, for the bifurcation diagram in Fig. 7, could one get oscillatory bursts if there was a slow variable driving the system between the dark-green and green regions, and non-oscillatory bursts if there was a slow variable driving the system between the dark-green and purple regions? If so, that would make a very nice story. And just to be clear, I'm emphasizing that they could use a story, but more simulations or analysis are not needed.

We agree with the reviewer that our spiking model cannot capture the spontaneous emission of population events through purely internal dynamics (without extreme fine tuning of model parameters). Rather, we assume an external, pulsatile input to a fraction of the network that can initiate a population event. In this slice experiments there could be a slow drifting excitability of a fraction of the neurons that would mimic such an event. Alternatively, as the reviewer conjectures, there may be slow and coordinated drift in the synaptic strength that would potentially initiate a population event. As requested we now address this possibility directly in the discussion at the top of pg. 24.

In our model the population events leave the low-activity fixed point through endogenous fluctuations modeled as Markov noise in the rate model, while in the network of spiking neuron models we employ an external brief 'kick' to 5% excitatory neurons. These perturbations are illustrative, and likely do not capture the complexity of the slice environment in experiment. Rather, they demonstrate that small, deviations from the low-activity state are sufficient to ignite full population events. Experimental and theoretical work shows several plausible sources of such deviations. First, spontaneous miniature synaptic events and asynchronous transmitter release create temporal fluctuations that can perturb a few neurons above threshold, seeding a network-wide burst (Sanchez-Vives, M.V. and McCormick, D.A., 2000.; Bazhenov, M., Timofeev, I., Steriade, M. and Sejnowski, T.J., 2002). These fluctuations arise from finite network size effects, stochastic vesicle release, and sporadic firing of highly excitable cells, effectively acting as the 'noise' term that displaces the network from rest. Second, slow modulatory processes can drift the synaptic depression parameters across bifurcation

boundaries, lowering the threshold for a noise-driven transition. While we lack of clear understanding of the underlying synaptic physiology in our slice experiments, our work explains that very subtle changes in the physiology can be important determinants of the structure of population events (without the depression mechanism, there would be no events, as illustrated by the dynamics in Fig. 10d).

Besides that, I have a few comments mainly related to clarity; the authors can ignore me if they think these are bad suggestions.

1. When analyzing the rate model, to what extent is stochasticity associated with the the birth-death model taken into account? My impression is that the conclusions about bifurcations are drawn from the deterministic model, and stochasticity is used to drive transitions. Is that correct? If so, it should be stated. Otherwise, it should be clear what's going on. (And I will freely admit that possibly they said this and I missed it.)

We agree with the reviewer that we should state how our bifurcation analysis is conducted in our Markov model. We now state it clearly in the bifurcation diagrams subsection in the Methods section.

Throughout the bifurcation analysis we consider the network model as deterministic. Finite-size stochasticity is introduced separately (Figs S1, 4) by treating the network as a birth–death Markov process to generate rare noise-driven excursions predicted by the deterministic phase portrait. We remark that even in our deterministic model the parameter N appears in the transfer f (see Eqs. (7) and (8)).

2. The description of what happens in the reduced (3-variable) models was very thorough. But I was missing why all the statements were true. For instance, on page 11, "Finally, for very large θ_{IE} the network remains bistable, but the saddle point is distant from the low activity state (Fig. 5b)."

Which saddle point? Is this the saddle point in the E-I phase plane? It would be nice if we had some pictures. Maybe nullclines at various values of p_{IE} ? Or maybe they can reduce the system to 2-D by considering fast inhibition, and then make phase-plane plots? The statements made vague sense to me given what I know about E-I network and depression, but it would be nice to get some more detailed intuition.

The same applies to the reduced model with p_{IE} fixed.

We agree with the reviewer that we need better clarification on how the bifurcation plots were constructed. However, since our reduced systems are still 3-dimensional then plotting phase portraits and nullcline structure may not bring deeper intuition. Nevertheless, we trust that the generic bifurcations in Figs. 5 and 6 (saddle-nodes and Hopfs) are familiar to the audience, at least in terms of the attractors that are involved (fixed points, saddles, and limit cycles). The first paragraph of the section entitled ‘Population event initiation’ on pg. 11 does introduce the various dynamical fixed points in our reduced (r_E, r_I, p_{IE}) system - with attention to the saddle point. We further add the following to pg. 11 when we begin our bifurcation analysis:

While this reduced model cannot replicate the population event dynamics of the full model, its simplicity can give insight on how events are initiated. We perform bifurcation analysis on the reduced system (r_E, r_I, p_{IE}) and use r_E as the indicator of network state (Figure 5b).

We have also added the following sentences to the Fig. 5 caption:

We remark that the bifurcation analysis is completed on the reduced 3-dimensional system (r_E, r_I, p_{IE}) , and we use the excitatory firing rate r_E as our indicator of the network dynamics.

and the Fig. 6 caption:

We remark that the bifurcation analysis is completed on the reduced 3-dimensional system (r_E, r_I, p_{EE}) , and we use the excitatory firing rate r_E as our indicator of the network dynamics.

3. Is Fig. 7 a cartoon, or quantitatively correct?

Fig. 7 is quantitatively correct.

4. What's the time constant for recovery of F and D (Eq 5) in the spiking model?

They are reported in the simulation parameters section now as $\tau^D = 103ms$ and $\tau^F = 96ms$

5. p. 20: What does "morally qualitatively" mean? Or is that a typo?

We thank the reviewer for pointing out the typo. We changed it to "**qualitatively analogous**" now.